# MILPnet: A Multi-Scale Architecture with Geometric Feature Sequence Representations for Advancing MILP Problems

**Ruobing Wang[1], Xin Li[1],[*] Mingzhong Wang[2]**
[1]Beijing Institute of Technology, Beijing, China
[2]University of the Sunshine Coast, Australia
{3120245684, xinli}@bit.edu.cn, mwang@usc.edu.au

## Abstract

We propose *MILPnet*, a multi-scale hybrid attention framework that models Mixed Integer Linear Programming (MILP) problems as geometric sequences rather than graphs. This approach directly addresses the challenge of Foldable MILP instances, a class of problems that graph-based models, specifically Graph Neural Networks (GNNs), fail to distinguish due to expressiveness limits imposed by the Weisfeiler-Lehman test. By representing MILPs through sequences of constraint and objective features, *MILPnet* captures both local and global geometric structure using a theoretically grounded multi-scale attention mechanism. We theoretically prove that *MILPnet* can approximate feasibility, optimal objective value, and optimal solution mappings over a measurable topological space with arbitrarily small error. Empirically, *MILPnet* outperforms graph-based methods in feasibility prediction accuracy and convergence speed on Foldable MILPs, while using significantly fewer parameters. It also generalizes effectively among problem scales and demonstrates strong performance on real-world MILP benchmarks when integrated into an end-to-end solver pipeline. The code is available at the link [1].

## 1 Introduction

Mixed-integer linear programming (MILP) is a foundational combinatorial optimization problem characterized by a linear objective function and linear constraints, with decision variables that can be either continuous or discrete. This flexibility makes MILP highly expressive and applicable in diverse theoretical and real-world domains, such as probabilistic optimal transport theory (Goldman & Trevisan, 2023) and transportation systems (Wang et al., 2023a), route optimization (Bula et al., 2016; Mammeri, 2019; Chen et al., 2023a), and power system planning (Zelaschi et al., 2025; Zhang et al., 2020).

However, as a well-known NP-hard problem, solving MILP remains a significant challenge. Traditional methods, such as Branch-and-Bound (Land & Doig, 1960) and Cutting Planes (Gomory, 2010), are commonly employed but become impractical for large-scale instances due to their intensive resource requirements.

Recently, Machine Learning methods have emerged as a promising alternative. ML models can approximate solutions efficiently by leveraging the implicit structure and patterns within MILP problems and integrating them with reinforcement learning or MILP Exact solvers. These approaches can significantly reduce computational costs while delivering effective solutions within practical timeframes (Bengio et al., 2020; Wu & Yang, 2022; Wang et al., 2024). Generative models, including diffusion models, have also been explored for solving structured MILP variants, such as the Traveling Salesman Problem (TSP) and the Maximum Independent Set Problem (MIS) (Sun & Yang, 2023; Sanokowski et al., 2024; Ma et al., 2024), showing strong performance and generalization.

---

[*]Corresponding author.
[1]https://github.com/ijklkjhggfajhhjkj/Seq_MILP.git

A recent trend has been to solve MILPs using Graph Neural Networks (GNNs), treating MILP instances as bipartite graphs that link variables and constraints (Han et al., 2023; Ye et al., 2023; Paulus & Krause, 2023; Geng et al., 2025; Liu et al., 2025). However, Bipartite graphs and GNNs can capture relationships between constraints and variables, missing the interactions between the constraints themselves, which potentially contain crucial features, such as feasible regions or optimal solutions of the MILP. Thus, GNN-based solutions suffer from a fundamental limitation: **they cannot distinguish between non-isomorphic MILP instances that differ in feasibility**, due to the expressive bounds of the Weisfeiler-Lehman (WL) test. As shown in recent work (Chen et al., 2023b), this leads to failure cases known as **Foldable MILP**s, where multiple distinct MILP instances are indistinguishable to GNNs but differ critically in their feasible regions.

Although, recent MILP representation research (Chen et al., 2023b) partially addresses this by injecting random features into graph structures. However, they only bypass WL-test limitations without capturing fundamental characteristics of MILP instances themselves. Current graph-based models remain inadequate for robust feasibility prediction and high-fidelity representation of general MILP problems, especially Foldable MILPs.

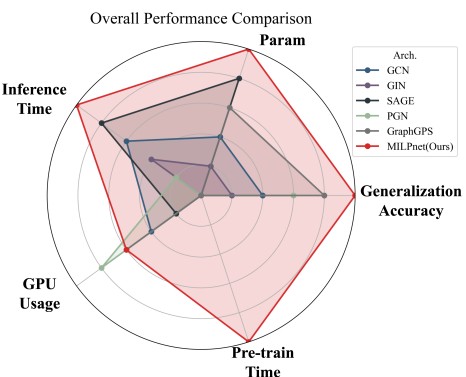

To overcome these limitations, we propose a novel representation framework that departs from the graph paradigm and instead adopts a computational geometry perspective. Considering the geometric and topological uniqueness of MILP problems (Huchette & Vielma, 2019; Conforti et al., 2010), we encode each MILP instance by extracting spatial geometric feature vectors, including hyperplane vectors from linear constraints, discrete integer point features, and direction vector from the linear objective function, and assemble these into a sequence. To the best of our

Figure 1: Overall performance comparison between MILPnet and graph-based models. **Larger area indicates better performance.**

knowledge, this is the first work to represent MILPs as sequences.

Given the complexity of MILP problems, we propose a Multi-Scale Hybrid Attention mechanism that enables our model, MILPnet, to learn both local and global features from the MILP sequence. Furthermore, we mathematically prove that our model can effectively approximate the feature mappings of any MILP instance with arbitrarily small error.

We validated our approach on Foldable MILP instances, which are specifically challenging for GNN- and Graph-based models. Across feasibility, optimal solution, and optimal object value predictions, MILPnet achieves great improvements of in accuracy and convergence speed, while using significantly fewer parameters and pre-training time. As summarized in Figure 1, MILPnet consistently outperforms baseline models across five key dimensions: generalization, accuracy, model size, inference time, and GPU memory usage. These results highlight the effectiveness of our geometric modeling and its broad potential.

## 2 RELATED WORK

MILP is a classic NP-hard problem (Karp, 1972). Traditional methods, such as branch-and-bound (Land & Doig, 1960) and cut-plane methods (Gomory, 2010), typically solve MILPs by simplifying or relaxing the problem. However, these methods can grow exponentially with the number of variables, making them computationally prohibitive for real-world or time-sensitive applications.

To address these challenges, machine learning methods have been applied to accelerate the MILP solving process(Bengio et al., 2020). Specifically, most methods model MILP instances with GNNs, framing them as weighted bipartite graphs (Gupta et al., 2020; Nair et al., 2021). These graphs typically consist of two disjoint sets of nodes: variables and constraints (Chen et al., 2023b). Message-passing mechanisms within GNNs are then used to capture their structural relationships. ML-based methods for MILP solving have been advanced by diffusion-model-based solvers (Sun & Yang, 2023;

Zhao et al., 2025; Sanokowski et al., 2024), predict and search solvers (Han et al., 2023; Huang et al., 2024), reinforcement learning methods (Wang et al., 2024; Feng & Yang, 2025), machine learning methods which reducing the problem solving complexity or combined with heuristic algorithms (He et al., 2014; Gasse et al., 2019; Chmiela et al., 2021; Paulus & Krause, 2023; Huang et al., 2023).

Formally, a bipartite graph is denoted as $G = (V \cup W, E)$, where $V$ and $W$ represent the variable and constraint nodes, respectively. The set $G_{m,n}$ includes all such graphs with $|V| = m$ and $|W| = n$, while the complete graph representation with node features for an MILP is given by $(G, H) \in G_{m,n} \times H_m^V \times H_n^W$.

While these graph-based models are effective in capturing variable-constraint relationships, they inherently miss higher-order interactions, particularly between constraints themselves, which may encode crucial information about feasibility or optimality. Therefore, for a class of MILP instances known as **Foldable MILP**s (Theorem 1), GNN-based models cannot represent their feasibility as the underlying WL-test cannot distinguish non-isomorphic graphs. Conversely, MILP instances that are not Foldable MILPs are Unfoldable MILPs.

**Theorem 1.** *(Lemma 3.2. in (Chen et al., 2023b)) There exist two MILP problems* $(G, H)$ *and* $(G, \hat{H})$, *with one being feasible and the other one being infeasible, such that* $(G, H) \sim (G, \hat{H})$.

This finding underscores a critical gap: graph-based models, regardless of feature augmentation, fundamentally cannot resolve the feasibility of Foldable MILP instances. It signifies the need for alternative representations beyond graph-based modeling to enhance the feasibility prediction for complex MILP problems.

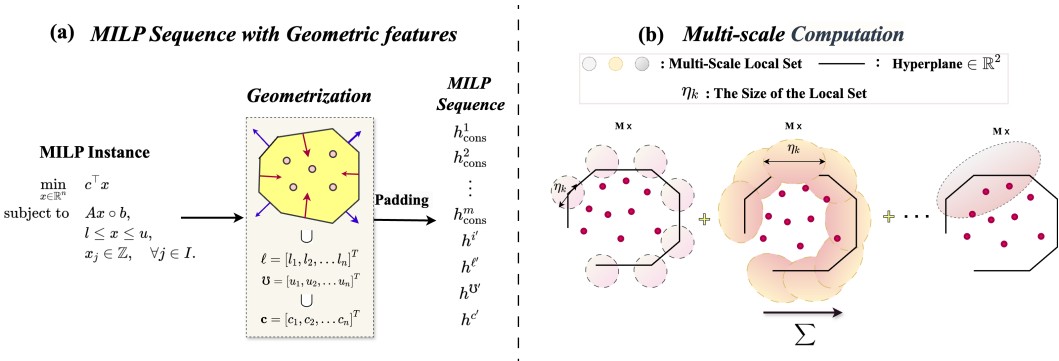

Figure 2: Overview of MILPnet. (a) An MILP instance is transformed into a sequence of geometric feature tokens, including constraint hyperplanes, variable bounds, integer indicators, and the objective vector. (b) The sequence is processed by a multi-scale hybrid attention architecture, enabling accurate approximation of feasibility, optimal objective value, and solution mappings.

## 3    PRELIMINARIES

**MILP formulation.** MILP is an NP-hard optimization problem characterized by a linear objective function and a set of linear constraints, with a subset of variables restricted to integer values. The standard formulation is:

$$\min_{x \in \mathbb{R}^n} c^\top x \quad \text{s.t.} \quad Ax \circ b, \quad l \le x \le u, \quad x_j \in \mathbb{Z} \, \forall j \in I \tag{1}$$

where $A \in \mathbb{R}^{m \times n}$, $b \in \mathbb{R}^m$, and $c \in \mathbb{R}^n$ are the problem parameters, $l, u \in (\mathbb{R} \cup \{\pm\infty\})^n$ specifies the variable bounds, $\circ \in \{\le, =, \ge\}^m$ denotes the constraint types, and $I \subseteq \{1, \dots, n\}$ indexes the variables that are required to be integer.

**Feasibility.** From a geometric perspective, define the continuous feasible region as a polyhedron $P = \{ x \in \mathbb{R}^n \mid A x \circ b, \, l \le x \le u \}$ and the integer lattice as $\mathbb{Z}_I^n = \{ x \in \mathbb{R}^n \mid x_j \in \mathbb{Z}, \, \forall j \in I \}$. The MILP feasible set is their intersection $S = P \cap \mathbb{Z}_I^n$, i.e. all points that satisfy the linear constraints and bounds while taking integer values in the specified dimensions. If $S = \emptyset$, the MILP is infeasible.

**Optimal solution and optimal objective value.** A point $x^* \in S$ is called an **optimal solution** if it minimizes the objective function over all feasible points: $c^\top x^* \leq c^\top x, \forall x \in S$. The corresponding scalar $c^\top x^*$ is the **optimal objective value**. If the objective can be decreased arbitrarily, i.e., for any $\epsilon > 0$, there exists $\hat{x} \in S$ such that $c^\top \hat{x} < -\epsilon$, the MILP is unbounded and the optimal value is $-\infty$.

# 4 SEQUENCE MODELING FOR MILP: MILP-SEQUENCE

This section introduces the first component of MILPnet (Figure 2 (a)), which encodes an MILP instance into a sequence representation from a geometric perspective. The MILP instance is decomposed into multiple components, each modeled in a well-defined topological space, forming what we refer to as the MILP-sequence.

## 4.1 GEOMETRIC MODELING OF MILP

We first reformulate the MILP problem from a geometric perspective. The MILP space consists of high-dimensional hyperplanes or half-spaces (from linear constraints and variable bounds), discrete point sets (from integrality requirements), and a direction vector (from the objective function). Together, these are represented as $P_{\text{constraint}} \cup P_{\text{range}} \cup I \cup c$. Each of these components is defined within a topological space as follows:

**Linear constraints.** The linear constraints are represented as the union of hyperplanes: $P_{\text{constraint}} = \bigcup_{i=1}^{m} H_i$, where $H_i$ is the $i$-th hyperplane or half-space defined by the vector $h_{cons}^i = (n_i, d_i, b_i)$, where $n_i \in \mathbb{R}^n$ is the normal vector of $H_i$, $d_i \in \{-1, 0, 1\}$ denotes its directional type, and $b_i \in \mathbb{R}$ is the bias term. $h_i$ is chosen from the topological space $H^{\text{cons}} = (\mathbb{R}^n) \times \{-1, 0, 1\} \times \mathbb{R}$.

**Variable bounds.** The bounds on variables define upper and lower half-spaces: $P_{\text{range}} = \left(\bigcup_{j=1}^{n} H_{\text{upper},j}\right) \bigcup \left(\bigcup_{j=1}^{n} H_{\text{lower},j}\right)$, where $H_{\text{upper},j} = \{x \in \mathbb{R}^n \mid x_j \leq u_j\}$ and $H_{\text{lower},j} = \{x \in \mathbb{R}^n \mid x_j \geq l_j\}$. We encode the bounds as vectors $h^\ell = (l_1, l_2, \ldots, l_n)$ and $h^\mho = (u_1, u_2, \ldots, u_n)$, drawn from the topological spaces $L = \prod_{j=1}^{n} L_i \subset \mathbb{R}^n$ and $U = \prod_{j=1}^{n} U_i \subset \mathbb{R}^n$, respectively, each equipped with the standard Euclidean topology. $\prod$ denotes the Cartesian product. The combined vector $\ell \cup \mho$ belongs to the space $H^{\text{Var}} = \mathbb{R}^{2n}$.

**Integer set.** The integrality constraints are encoded using a binary vector: $h^i = (i_1, i_2, i_3, \ldots, i_n) \in \{0, 1\}^n$, where $i_k = 1$ if $x_k$ is constrained to be an integer, and $i_k = 0$ otherwise. This vector resides in the discrete topological space $I = \{0, 1\}^n$.

**Linear optimization direction.** The objective function is represented by a coefficient vector $h^c = (c_1, c_2, \ldots, c_n) \in \mathbb{R}^n$, drawn from the topological space $H^{\text{obj}} = \mathbb{R}^n$.

It is worth noting that, although the MILP feasible region is mathematically defined by intersections, we use unions in our geometric reformulation to denote the collection of individual components (e.g., constraint hyperplanes and bounds) treated as sequence tokens. This formulation expands the representational space, enabling MILPnet to more effectively explore the solution structure and approximate optimal outcomes.

## 4.2 MEASURES OF THE MILP GEOMETRIC SPACES

To unify component representations and manage dimensionality differences, we apply zero-padding, yielding two related topological spaces $H^{\text{MILP0}}$ (original) and $H^{\text{MILP}}$ (padded). The padded component spaces include $H^{\text{cons}}, H^{\text{Var}'}, H^{\text{obj}'}, I'$. As shown in Theorem 5 in Appendix, the spaces padded and non-padded are homeomorphic, ensuring no representation loss.

**Measures.** We equip each continuous space $(\mathbb{R}^n, \mathbb{R}^{2n}, \mathbb{R})$ with Borel $\sigma$-algebra and Lebesgue measure, and each discrete component $(\{-1, 0, 1\}, \{0\}, \{0, 1\}^n)$ with the counting measure. By product construction, the component measures are $\mu_{H^{\text{cons}}} = \lambda_{\mathbb{R}^n} \times \mu_{\{-1,0,1\}} \times \lambda_{\mathbb{R}}$, $\mu_{H^{\text{Var}'}} = \lambda_{\mathbb{R}^{2n}} \times \mu_{\{0\}}^4$, $\mu_{H^{\text{obj}'}} = \lambda_{\mathbb{R}^n} \times \mu_{\{0\}}^2$, $\mu_{I'} = \mu_{\{0,1\}^n} \times \mu_{\{0\}}^2$. The overall padded topological space is $H^{\text{MILP}} = ((H^{\text{cons}})^m \cup H^{\text{Var}'}) \times I' \times H^{\text{obj}'}$, which is measurable by construction.

### 4.3 Geometric Feature Integration

We now integrate the MILP geometric feature vectors into a sequence-based representation, treating each feature vector as a token.

**MILP-sequence**. Since the linear constraints $h_{cons}^i$, integer point sets $h^{i'}$, and variable ranges $h^{\ell'}$ and $h^{\mho'}$ are permutation invariant (as formalized in Theorems 6 and 7 in Appendix), we can arrange them in any order. To preserve this invariance and recognize the term objective function's unique role, we place the objective token $h^{c'}$ at the end of the sequence. Thus, the MILP-sequence is:

$$\mathbf{x} = [h_{\text{cons}}^1, h_{\text{cons}}^2, \dots, h_{\text{cons}}^m, h^{i'}, h^{\ell'}, h^{\mho'}, h^{c'}], \quad h \in \mathbb{R}^{n+2} \tag{2}$$

The topological space corresponding to the MILP-sequence is defined as $H^{\text{MILP}} \subset \mathbb{R}^{(m+4)(n+2)}$.

**Sequence-based mappings**. With this representation, we define the core mappings used to analyze MILP instances: **Feasibility mapping** $\Phi_{\text{feas}} : H^{\text{MILP}} \to \{0, 1\}$, **Optimal objective value mapping** $\Phi_{\text{obj}} : H^{\text{MILP}} \to \mathbb{R} \cup \{\infty, -\infty\}$, and **Optimal solution mapping** $\widetilde{H}^{\text{MILP}} \cap \Phi_{\text{feas}}^{-1}(1) \to \mathbb{R}^n$. These mappings are formally defined in Definitions 8 to 10 in the Appendix. We prove their measurability in Appendix G.

## 5 Multi-Scale Hybrid Attention for MILP-sequence

This section introduces the second core component of MILPnet (Figure 2 (b)): a novel Multi-Scale Hybrid Attention mechanism specifically designed to model MILP sequences. This architecture enables the network to capture both fine-grained local structure and global context by combining multiple levels of attention. It operates directly on the MILP-sequence defined in the previous section and is supported by a rigorous approximation theory over the measurable space $H^{\text{MILP}}$.

### 5.1 Shifted-Window Multi-Scale Attention

The geometric structure of MILPs suggests that relationships among constraints can reflect topological characteristics of the feasible region. To leverage this structure, we employ multi-scale local attention via sliding windows that extract features from various neighborhoods in the MILP-sequence.

**Shifted-Window Local Attention.** Given an embedded MILP-sequence $\mathbf{X}_{\text{embed}} \in \mathbb{R}^{(m+4) \times (n+2)}$, derived through linear projection, we define local attention windows that slide across the sequence. For each token at position $i \in \{1, \dots, m+4\}$, a window of size $\eta_k$ is centered at $i$, covering elements from position $i_{\min} = \max(i - \lfloor \frac{\eta_k - 1}{2} \rfloor, 1)$ to position $i_{\max} = \min(i + \lceil \frac{\eta_k - 1}{2} \rceil, m+4)$. To effectively capture the relationship between multi-level features and the MILP's overall goal, this window also needs to incorporate the embedding of the objective function feature $h^{c'}$ at position $m+4$. Therefore, the position indices $\eta_k(i)$ for the window attention at position $i$ with the window size $\eta_k$ are defined as:

$$\eta_k(i) = \{j \in [i_{\min}, i_{\max}]\} \cup \{m+4\}. \tag{3}$$

Then, the local attention at position $i$ for scale $\eta_k$ is computed as:

$$\mathbf{Q}_i^{\eta_k} = \mathbf{W}_Q \mathbf{X}_{\text{embed},i}, \mathbf{K}_j^{\eta_k} = \mathbf{W}_K \mathbf{X}_{\text{embed},j}, \mathbf{V}_j^{\eta_k} = \mathbf{W}_V \mathbf{X}_{\text{embed},j}, \quad j \in \eta_k(i) \tag{4}$$

$$\alpha_{ij}^{\eta_k} = \text{softmax}\left(\frac{\mathbf{Q}_i^{\eta_k} \cdot (\mathbf{K}_j^{\eta_k})^\top}{\sqrt{d_k}}\right), \quad j \in \eta_k(i), \quad \mathbf{Att}_i^{\eta_k} = \sum_{j \in \eta_k(i)} \alpha_{ij}^{\eta_k} \mathbf{V}_j^{\eta_k} \tag{5}$$

where $\mathbf{W}_Q$, $\mathbf{W}_K$, $\mathbf{W}_V$ are the linear transformation matrices for Query, Key, and Value, and $d_k$ is the dimension of the key vectors, which is split from the embedding size $d_{dim}$ by the multi-heads. softmax function is used to normalize the Attention weights.

**Multi-Scale MILP-sequence Attention.** To aggregate information at different granularities, we apply local attention using $N$ window sizes. The resulting outputs are averaged to produce the multi-scale representation,which is like the style in the Figure 2 (b),i.

$$\mathbf{Att}^{\text{multi}} = \frac{1}{N} \sum_{i=1}^{m+4} \sum_{k=1}^{N} \mathbf{Att}_i^{\eta_k} = \frac{1}{N} \sum_{k=1}^{N} \sum_{i=1}^{m+4} \left( \sum_{j \in \eta_k(i)} \alpha_{ij}^{\eta_k} \mathbf{V}_j^{\eta_k} \right) \tag{6}$$

where window sizes $\eta_k \in [2, m + 4]$ allow for a comprehensive evaluation of local and global contextual influences, the summation over $i$ represents concatenation along the sequence dimension. Let $\eta_{\max} = \max_k(\eta_k)$ denote the largest window size.

**Hybrid Attention Integration.** To capture both multi-scale locality and global context, we define a hybrid attention mechanism that integrates multi-scale attention with global attention using a learnable parameter $\alpha$:

$$\mathbf{Att}^{\text{hybrid}} = \alpha \cdot \mathbf{Att}^{\text{multi}} + (1 - \alpha) \cdot \mathbf{Att}^{\text{global}} \tag{7}$$

As illustrated in Figure 2, each MILP instance is first geometrically encoded as a feature vector in $\mathbb{R}^{n+2}$, and then padded to form an MILP-sequence of length $m + 4$. This sequence undergoes linear projection and position encoding to produce the embedded input: $\mathbf{z} = [\mathbf{z}_1, \mathbf{z}_2, ..., \mathbf{z}_{n+2}] + \mathbf{z}_{\text{position}}$. With this embedded sequence, MILPnet applies its Hybrid Attention (HYA) mechanism, followed by residual connections and layer normalization. The computation at the $l$-th layer proceeds as:

$$\hat{\mathbf{z}}^l = \text{HYA}(\mathbf{z}^{l-1}) + \mathbf{z}^{l-1}, \mathbf{z}^l = \text{MLP}(\text{LN}(\hat{\mathbf{z}}^l)) + \hat{\mathbf{z}}^l, \mathbf{z}^l = \text{LN}(\mathbf{z}^l) \tag{8}$$

where LN denotes layer norm, MLP refers to a position-wise feedforward network. More detailed information is in Figure 5 from Appendix.

**Time Complexity.** The computational complexity of the hybrid attention is $O\big(h\, d\, (m+4)^2\, (N+1)\big)$ where $h$ is the number of attention heads, $d$ is the embedding size, $m$ is the number of the constraints, and $N$ is the number of windows. Detailed analysis is provided in Appendix I.

## 5.2 MILPNET REPRESENTATION ON THE MEASURABLE SPACE $H^{\text{MILP}}$

We now formally demonstrate that the multi-scale sequence architecture of MILPnet is capable of approximating the feature mappings of any MILP instance, when represented as a sequence in the measurable topological space $H^{\text{MILP}}$. We define two function classes: $\mathcal{F}_{\text{HYA}}^{\text{MILPnet}} : H^{\text{MILP}} \to \mathbb{R}$ for scalar-valued network mappings, and $\mathcal{F}_{\text{HYA,V}}^{\text{MILPnet}} : H^{\text{MILP}} \to \mathbb{R}^n$ for vector-valued network mappings with fixed output dimension $n$. By leveraging these mappings and the measurability structure introduced in Section 4.2 and 4.3, we can prove that for any MILP instance viewed as a sequence:

> *MILPnet can uniformly approximate MILP feasibility mapping, MILP optimal-solution mapping, and MILP optimal objective value mapping.*

The following theorems formally establish MILPnet's approximation capabilities (see Appendix G for detailed proofs and corollary on infinite set):

**Theorem 2.** *Let $D \subset H^{MILP}$ be a finite dataset. For any $\epsilon > 0$, there exists a neural network $F_{HYA} \in \mathcal{F}_{HYA}^{MILPnet}$ such that:*

$$P\left(\mathbb{I}_{F_{HYA}(x) > \frac{1}{2}} \neq \Phi_{feas}(x)\right) < \epsilon, \quad \forall x \in D, \tag{9}$$

**Theorem 3.** *Let $D \subset H^{MILP}$ be a finite dataset. For any $\epsilon, \delta > 0$, there exist two neural networks $F_{HYA,1}, F_{HYA,2} \in \mathcal{F}_{HYA}^{MILPnet}$ such that for classifying whether the objective value is finite:*

$$P\left(\mathbb{I}_{F_{HYA,1}(x) > \frac{1}{2}} \neq \mathbb{I}_{\Phi_{obj}(x) \in \mathbb{R}}\right) < \epsilon, \quad \forall x \in D \tag{10}$$

*where $\mathbb{I}_{\Phi_{obj}(x) \in \mathbb{R}}$ is an indicator function that determines whether the objective value is finite. And for the regression problem of predicting the objective value:*

$$P\left(|F_{HYA,2}(x) - \Phi_{obj}(x)| > \delta\right) < \epsilon, \quad \forall x \in D \cap \Phi_{obj}^{-1}(\mathbb{R}) \tag{11}$$

**Theorem 4.** *Let $D \subset \Phi_{obj}^{-1}(\mathbb{R}) \subset H^{MILP}$ be a finite dataset. For any $\epsilon, \delta > 0$, there exists a Hybrid attention based network $F_{HYA,V} \in \mathcal{F}_{HYA,V}^{MILPnet}$ such that:*

$$P\left(\|F_{HYA,V}(x) - \Phi_{solu}(x)\| > \delta\right) < \epsilon, \quad \forall x \in D, \tag{12}$$

To train any MILPnet $F_\phi$ to approximate these mappings, we minimize the error between MILPnet and the feature mapping as the loss function $\mathcal{L}(\phi) = E[\|\mathbf{y} - F_\phi(x)\|_2]$, where $\mathbf{y}$ is ground truth.

Table 1: Generalization results for feasibility mapping on FOLD(20,*) and FOLD(50,*), with 10,000 foldable instances per setting and pre-training times of 3 and 10 minutes. Full results are provided in Appendix B.5.

| Method | Type | FOLD(20,6) | | | FOLD(20,16) | | | FOLD(50,20) | | | FOLD(50,30) | | |
|---|---|---|---|---|---|---|---|---|---|---|---|---|---|
| | | MSE | ErrorN | Params | MSE | ErrorN | Params | MSE | ErrorN | Params | MSE | ErrorN | Params |
| SCIP | Exact | —— | 0 | —— | —— | 0 | —— | —— | 0 | —— | —— | 0 | —— |
| GCN | Graph | 0.3073 | 5000 | 1.21M | 0.3073 | 5000 | 1.21M | 0.3583 | 5000 | 1.21M | 0.3556 | 5000 | 1.21M |
| GIN | Graph | 0.4200 | 5000 | 1.63M | 0.4209 | 5000 | 1.63M | 0.4200 | 5000 | 1.63M | 0.4204 | 5000 | 1.63M |
| SAGE | Graph | 0.4642 | 5000 | 0.66M | 0.4999 | 5000 | 0.66M | 0.4714 | 5000 | 0.66M | 0.4586 | 5000 | 0.66M |
| PGN | Graph | 0.2508 | 5000 | 1.64M | 0.2523 | 5000 | 1.64M | 0.2511 | 5000 | 1.64M | 0.2511 | 5000 | 1.64M |
| GraphGPS | Graph | 0.2500 | 5000 | 0.66M | 0.2500 | 5000 | 0.66M | 0.2500 | 5000 | 0.66M | 0.2500 | 5000 | 0.66M |
| GCN$^{rf}$ | Rf Graph | 0.2498 | 5000 | 1.21M | 0.2500 | 5000 | 1.21M | 0.2476 | 4334 | 1.21M | 0.5223 | 5000 | 1.21M |
| GIN$^{rf}$ | Rf Graph | 0.2500 | 5000 | 1.63M | 0.2501 | 5000 | 1.63M | 0.2500 | 5000 | 1.63M | 0.4204 | 5000 | 1.63M |
| SAGE$^{rf}$ | Rf Graph | 0.2499 | 5009 | 0.66M | 0.2499 | 4995 | 0.66M | 0.2500 | 4997 | 0.66M | 0.2500 | 5002 | 0.66M |
| PGN$^{rf}$ | Rf Graph | 0.2582 | 5000 | 1.64M | 0.2560 | 5000 | 1.64M | 0.2502 | 5000 | 1.64M | 0.2502 | 5000 | 1.64M |
| GraphGPS$^{rf}$ | Rf Graph | 0.2510 | 5000 | 0.66M | 0.2510 | 5000 | 0.66M | 0.2500 | 5000 | 0.66M | 0.2502 | 5000 | 0.66M |
| MILPnet | Ours (Seq) | **0.0005** | **0** | **0.56M** | **0.0004** | **0** | **0.56M** | **0.0005** | **0** | **0.60M** | **0.0023** | **12** | **0.60M** |

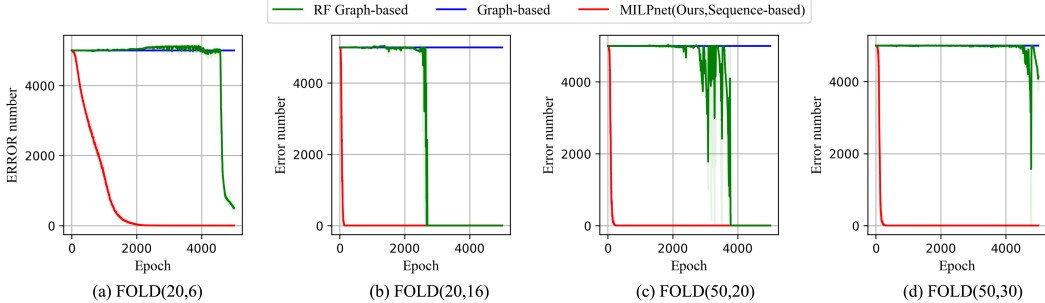

Figure 3: Representation learning Dynamics on feasibility for FOLD20 and FOLD50. Full results are in Figure 8 in Appendix.

# 6 EXPERIMENT

We conducted comprehensive experiments to evaluate the effectiveness, efficiency, and generalizability of MILPnet to answer:**RQ1:** How effectively does MILPnet represent and generalize on Foldable MILP instances? **RQ2:** How does MILPnet perform on real-world MILP instances? **RQ3:** How do MILPnet's architectural components impact performance?

## 6.1 EXPERIMENT SETUP

**Datasets.** We evaluate MILPnet on two categories of datasets: (1) *Synthetic Foldable MILP Instances (FOLD$(n, m)$)*: Following (Chen et al., 2023b), we generate Foldable MILP instances where $n$ and $m$ are the number of variables and constraints, respectively. (2) *Real-World MILP Benchmarks*: We select four common MILP benchmarks: IP (Item Placement) from ML4CO competition dataset (Gasse et al., 2022), SC (Feige, 1998; Chvátal, 1979), CA (Leyton-Brown et al., 2000), and FC (Charikar et al., 1999). They are all Unfoldable MILP instances in Classical ML4CO paper (Gasse et al., 2019). More details on the datasets are provided in Appendix K.3 and Appendix K.5.

**Baselines.** We compare MILPnet against several representative graph-based learning models, including GCN (Chen et al., 2023b),GIN (Xu et al., 2019),SAGE (Wu et al., 2021),PGN (Cappart et al., 2022; Veličković et al., 2020),GraphGPS (Wang et al., 2023b). Each baseline has an augmented variation with random features (RF), indicated by a superscript $*^{rf}$, specifically designed to overcome WL-test limitations in feasibility prediction. Appendix K.1 provides more details on the baselines.

**Metrics.** We use *MSE* (Mean Squared Error) and *ErrorN* (number of incorrect predictions) (Chen et al., 2023b), *Params* (number of model parameters) to evaluate performance. Lower is better.

Table 2: Generalization experiments for *End-to-End* optimal solution predict 1-hour of pre-train.

| Method | Type | FOLD(20,6) | | FOLD(20,16) | |
|---|---|---|---|---|---|
| | | MSE | Params | MSE | Params |
| GCN | Graph | 0.0751 | 1.17M | 0.2000 | 1.17M |
| GIN | Graph | 0.0753 | 1.59M | 0.2000 | 1.59M |
| SAGE | Graph | 0.0750 | 0.60M | 0.2000 | 0.60M |
| PGN | Graph | 70.1485 | 1.64M | 20.5955 | 1.64M |
| GraphGPS | Graph | 0.0863 | 0.60M | 4.0395 | 0.60M |
| MILPnet(Ours) | Sequence | **0.0473** | 0.56M | **0.1964** | 0.56M |
| Method | Type | FOLD(50,20) | | FOLD(50,30) | |
| | | MSE | Params | MSE | Params |
| GCN | Graph | **0.1000** | 1.17M | 0.1509 | 1.17M |
| GIN | Graph | 0.1000 | 1.59M | 0.1501 | 1.59M |
| SAGE | Graph | 0.1000 | 0.86M | **0.1500** | 0.86M |
| PGN | Graph | 55.8530 | 1.64M | 36.5160 | 1.64M |
| GraphGPS | Graph | 0.1020 | 0.67M | 0.1551 | 0.67M |
| MILPnet(Ours) | Sequence | 0.1158 | 0.62M | 0.1654 | 0.63M |

Table 3: Generalization experiments for *End-to-End* optimal objective value prediction

| Method | Type | FOLD(20,6) | | FOLD(20,16) | |
|---|---|---|---|---|---|
| | | MSE | Params | MSE | Params |
| GCN | Graph | 1.776e-10 | 0.88M | 3.928e-10 | 0.58M |
| GIN | Graph | 6.489e-10 | 0.12M | 6.489e-10 | 0.16M |
| SAGE | Graph | 2.235e-9 | 0.66M | 6.742e-10 | 0.60M |
| PGN | Graph | 1.958e-10 | 2.91M | 5.4934e-10 | 2.91M |
| GraphGPS | Graph | 3.769e-6 | 0.92M | 1.365e-05 | 0.92M |
| MILPnet(Ours) | Sequence | **1.309e-10** | 0.56M | **2.828e-10** | 0.56M |
| Method | Type | FOLD(50,20) | | FOLD(50,30) | |
| | | MSE | Params | MSE | Params |
| GCN | Graph | 3.333e-10 | 0.96M | 5.8012e-10 | 0.96M |
| GIN | Graph | 2.629e-10 | 1.30M | 3.458e-10 | 1.47M |
| SAGE | Graph | 1.219e-8 | 0.66M | 7.555e-9 | 0.92M |
| PGN | Graph | 3.935e-10 | 2.91M | 3.762e-9 | 2.30M |
| GraphGPS | Graph | 2.531e-7 | 0.92M | 3.442e-6 | 0.92M |
| MILPnet(Ours) | Sequence | **9.458e-12** | 0.60M | **3.007e-10** | 0.60M |

Table 4: Generalization results for feasibility prediction on larger Foldable instances FOLD(200,20), FOLD(300,40), and FOLD(500,60), each with 10,000 instances and 1 hour of pre-training.

| Method | Type | FOLD(200,20) | | | FOLD(300,40) | | | FOLD(500,60) | | |
|---|---|---|---|---|---|---|---|---|---|---|
| | | MSE | ErrorN | Params | MSE | ErrorN | Params | MSE | ErrorN | Params |
| SCIP | Exact | —— | 0 | —— | —— | 0 | —— | —— | 0 | —— |
| GCN | Graph | 0.2676 | 5000 | 0.03M | 0.3073 | 5000 | 0.08M | 0.2500 | 4999 | 0.14M |
| GIN | Graph | 0.2573 | 5000 | 0.04M | 0.3099 | 5000 | 0.10M | 0.2500 | 5000 | 0.21M |
| SAGE | Graph | 0.2814 | 5000 | 0.03M | 0.3951 | 5000 | 0.05M | 0.2500 | 5000 | 0.12M |
| PGN | Graph | 0.2508 | 5000 | 0.08M | 0.2523 | 5000 | 0.14M | 0.2523 | 5000 | 0.29M |
| GraphGPS | Graph | 0.2500 | 5000 | 0.03M | 0.2500 | 5000 | 0.59M | 0.2501 | 5000 | 0.92M |
| GCN[rf] | Rf Graph | 0.2497 | 4835 | 0.03M | 0.2500 | 5000 | 0.80M | 0.2611 | 4999 | 0.14M |
| GIN[rf] | Rf Graph | 0.2500 | 4998 | 0.04M | 0.2500 | 5012 | 0.10M | 0.2501 | 5003 | 0.21M |
| SAGE[rf] | Rf Graph | 0.2500 | 5011 | 0.03M | 0.2500 | 4998 | 0.05M | 0.2500 | 4981 | 0.12M |
| PGN[rf] | Rf Graph | 0.2544 | 5000 | 0.08M | 0.2500 | 5000 | 0.14M | 0.2506 | 5000 | 0.29M |
| GraphGPS[rf] | Rf Graph | 0.2502 | 5000 | 0.03M | 0.2510 | 5000 | 0.05M | 0.2506 | 5000 | 0.92M |
| MILPnet | Seq (Ours) | **0.0155** | **191** | **0.02M** | **0.0521** | **560** | **0.05M** | **0.1832** | **2453** | **0.11M** |

## 6.2 PERFORMANCE ON FOLDABLE MILP INSTANCES

**Representation Effectiveness.** We evaluate *MILPnet* on Foldable MILP instances with increasing complexity: FOLD(20,6), FOLD(20,16), FOLD(50,20), and FOLD(50,30). Figure 3 shows that *MILPnet* converges rapidly to near-zero *ErrorN* while standard graph-based models fail to improve beyond their initial performance. Models with random feature augmentation show moderate improvements on simpler instances but struggle to converge on more complex cases such as FOLD(50,30). Similar trends are observed in MSE (more details in Appendix B.4). These results confirm that *MILPnet* effectively captures the geometric and combinatorial structure of MILPs, providing empirical support for the theoretical guarantees in Theorems 2–4.

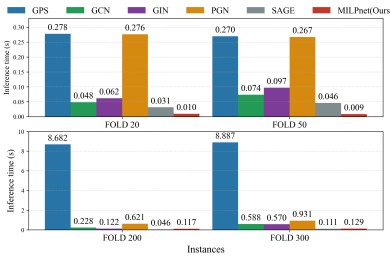

Figure 4: Inference time comparisons

**Generalization Abilities.** We evaluate *MILPnet* on both small (FOLD(20,*), FOLD(50,*)) and large (FOLD(200,20), FOLD(300,40), and FOLD(500,60)) Foldable MILP instances under equal pre-training time, focusing on three *End-to-End* prediction tasks: feasibility, optimal solution, and optimal objective value. In feasibility prediction (Tables 1 and 4), *MILPnet* improves MSE by 1–3 orders of magnitude and reduces *ErrorN* by 90% or more in most cases. For optimal solution and value prediction (Tables 2 and 3), *MILPnet* exceeds baselines while using substantially fewer parameters.

**Efficiency Analysis and Sparse Variant.** Figure 4 shows that *MILPnet* consistently achieves the fastest inference time across both small and large Foldable MILPs. GPU memory usage (Appendix Figure 12) remains moderate (< 4GB on FOLD300), making it suitable for deployment on a wide range of hardware. We also provide a sparse (stride) MILPnet variant that reduces the time complexity to $\frac{1}{s}$ of the original, with *FOLD*200 faster inference results shown in Table 14 and details in the AppendixJ,indicates its potential computational scalability.

Table 5: *Predict + Search* performance of MILPnet on four benchmarks. "Rand" denotes randomly generated initial solutions. "gap" is computed as gap $= (\hat{y} - y^*) / (|y^*| + 10^{-9})$, where $y^*$ is the exact optimal value by SCIP. "PT" means the predicting optimal candidate solution. "ST" is total solving time including search. Full Comparisons with advanced methods are in Appendix C.1.

| Method | Type | IP | | | SC | | | CA | | | FC | | |
|---|---|---|---|---|---|---|---|---|---|---|---|---|---|
| | | gap($\downarrow$) | PT(s) ($\downarrow$) | ST(s) ($\downarrow$) | gap($\downarrow$) | PT(s) ($\downarrow$) | ST(s) ($\downarrow$) | gap($\downarrow$) | PT(s) ($\downarrow$) | ST(s) ($\downarrow$) | gap($\downarrow$) | PT(s) ($\downarrow$) | ST(s) ($\downarrow$) |
| Rand | Random | 36.5397 | —— | 0.1240 | 1.7166 | —— | 2.0173 | 2.8057 | —— | 6.7830 | 1.4939 | —— | 1.7786 |
| GCN | Graph | 0.0389 | 0.2443 | 0.3774 | 1.1461 | 0.2391 | 0.8386 | 0.9890 | 0.2278 | 3.2328 | 0.4257 | 0.2518 | 1.4875 |
| MILPnet | Seq(Ours) | **0.0234** | **0.0625** | **0.1864** | **0.3483** | **0.0137** | **0.6915** | **0.7651** | **0.0139** | **3.2300** | **0.3503** | **0.0177** | **1.3773** |

Table 6: Performance impact of MILPnet's architectural components

(a) Ablation results on the impact of HYA and MSA

| Method | FOLD(20,6) | | FOLD(50,20) | | FOLD(100,20) | |
|---|---|---|---|---|---|---|
| | MSE | ErrorN | MSE | ErrorN | MSE | ErrorN |
| MILPnet (ours) | **0.0001** | **0** | **0.0001** | **0** | **0.0074** | **6** |
| w/o HYA | 0.2503 | 5036 | 0.2501 | 4999 | 0.2501 | 4987 |
| w/o MSA | 0.2500 | 5051 | 0.2500 | 5008 | 0.2500 | 4995 |

(b) Impact of $\eta_{\max}$ ("/$n$" means $\eta_{\max} = n$)

| Method | FOLD(20,6) | | FOLD(50,30) | | FOLD(100,20) | |
|---|---|---|---|---|---|---|
| | MSE | ErrorN | MSE | ErrorN | MSE | ErrorN |
| MILPnet /2 | **0.0005** | **0** | **0.0477** | **235** | 0.0558 | 318 |
| MILPnet /3 | 0.0006 | 0 | 0.0481 | 259 | **0.0471** | **272** |
| MILPnet /4 | 0.0026 | 0 | 0.0841 | 764 | 0.0774 | 727 |

**Overall Model Comparison.** We summarize performance across key metrics, including generalization, parameter count, inference time, GPU memory, and *ErrorN*, using radar charts with performance ranks(Figures 1 for FOLD50 and Appendix Figure 13 for FOLD300). In both cases, *MILPnet* consistently forms the outermost polygon, indicating superior performance in accuracy, efficiency, and resource usage compared to graph-based baselines.

## 6.3 PERFORMANCE ON REAL-WORLD MILP INSTANCES

While our focus is on sequence-based representation, we further demonstrate the utility of *MILPnet* in solving real-world MILP problems. We adopt a *predict + search* approach, where the model is trained to predict an optimal solution, which is then refined via a lightweight local heuristic. Table 5 compares MILPnet with graph-based baselines and a random search baseline. *MILPnet* consistently achieves the smallest optimality gap to the exact solver, and the lowest overall solving time. These results demonstrate the model's strong representation quality and practical solving effectiveness.

## 6.4 IMPACT OF ARCHITECTURAL COMPONENTS

**Ablation Study on Hybrid and Multi-scale Attention**. To evaluate the contribution of MILPnet's attention design, we compare the removal of the Hybrid Attention (HYA) and Multi-Scale Attention (MSA) components. Table 6a shows that removing either component significantly degrades feasibility prediction accuracy. In particular, models without MSA struggle to capture local structure, while disabling HYA weakens global context integration. These results confirm that both components are essential to MILPnet's representation effectiveness.

Table 7: Permutation invariance Experiments. "Or *" represents randomly permuted constraint order in the MILP-sequence."Original *"represents the original order.

| Order | Method + Arch | FOLD(50,20) | | FOLD(50,30) | |
|---|---|---|---|---|---|
| | | MSE | ErrorN | MSE | ErrorN |
| GCN (Original) | GNN + Graph | 0.4719 | 5000 | 0.3242 | 5000 |
| Original | MILPnet + Seq | 0.0005 | 0 | 0.0022 | 0 |
| Or 1 | MILPnet + Seq | 0.0006 | 0 | 0.0021 | 0 |
| Or 2 | MILPnet + Seq | 0.0001 | 0 | 0.0014 | 0 |
| Or 3 | MILPnet + Seq | 0.0004 | 0 | 0.0021 | 9 |

**Ablation study on multi-scale attention blocks** We conduct an ablation study on the depth of the multi-scale attention module by varying the number of blocks from 1 to 3. The results, which visualize and summarize its representational and generalization capabilities, are provided in Figure 7 in the Appendix B and Table 10. A key finding is that deeper multi-scale attention modules offer a significant improvement in convergence speed and representation at no cost to generalization ability.

**Impact of the Maximum Window Size** As shown in Tables 6b (pre-train 5mins) and Appendix Fig 6, the maximum window size $\eta_{\max}$ influences the trade-off between convergence speed and representational capacity. Smaller windows accelerate convergence on simpler instances like FOLD(20,), while large windows slow training and degrade performance on complex cases such as FOLD(100,20). These results indicate that $\eta_{\max}$ requires careful tuning and does not follow a monotonic pattern.

**Permutation Sensitivity of MILP-sequence.** Randomly shuffling the constraint and variable order in FOLD 20 and FOLD 50 keeps retrained MILPnet's MSE within the same magnitude (Table 7,11 in Appendix B) and well below that of other models (typical MSE $\approx$ 0.x, error $\approx$ 5000), for three seeds. These results demonstrate that ours MILP-sequence construction preserves invariance under both constraint and variable permutations in this scale for long time pre-training.

## 6.5 MILPNET SOLVING EFFICIENCY ANALYSIS

We analyze the solving efficiency of MILPnet across two dimensions: **(i)** Large-scale public MILP benchmarks (with 1000 variables, details in the Appendix) to evaluate each framework component; **(ii)** Heterogeneous generalization and scalability on four very large real-world benchmarks (with 10000+ variables). To quantify solution quality in details, we integrate both MILPNet and graph-based baselines as pre-solving method(eg. (Nair et al., 2021)) like for predicting candidate solutions by learning from MSE, and use the refined solutions by fast heuristic as the warm-start solution for branch-and-bound framework in **(i)**.

**Impact of Window Size $\eta_{\mathbf{max}}$ and Attention Depth $L$** We evaluate the impact of maximum window size $\eta_{\max}$ and attention depth $L$ on two large MILP benchmarks. Table 8 show that performance is relatively insensitive to these hyperparameters, with consistent improvements (*Nodes, Branching Time, Dual Gap*) over graph-based methods and Strong Branching across all configurations.

Table 8: *Branch and Bound* performance of MILPnet (Neural pre-solving with fast heuristic: H.Seq.) on 1000+ variable benchmarks for 50 instances within 60s solving limit.

| Method | Type | SC(1000,500) | | | | CA(1000,500) | | | |
|---|---|---|---|---|---|---|---|---|---|
| | | Obj.($\downarrow$) | Node($\downarrow$) | Time ($\downarrow$) | Dual Gap($\downarrow$) | Obj.($\uparrow$) | Node($\downarrow$) | Time ($\downarrow$) | Dual Gap($\downarrow$) |
| SB | Exact | $230.50 \pm 28.27$ | $39.0 \pm 41.2$ | $19.1 \pm 15.0$ | $0.21\pm 1.23(\%)$ | $146.0 \pm 26.6$ | $1.0 \pm 0.0$ | $3.4 \pm 1.0$ | $0.00\pm0.00(\%)$ |
| GIN | H. Graph | $230.56 \pm 28.35$ | $38.8 \pm 40.0$ | $18.6 \pm 15.4$ | $0.21\pm 1.23(\%)$ | $146.0 \pm 26.6$ | $1.0 \pm 0.0$ | $1.8 \pm 0.2$ | $0.00\pm0.00(\%)$ |
| GraphGPS | H. Graph | $230.56 \pm 28.35$ | $38.9 \pm 44.8$ | $17.9 \pm 15.1$ | $0.21\pm 1.23(\%)$ | $146.0 \pm 26.6$ | $1.0 \pm 0.0$ | $1.8 \pm 0.2$ | $0.00\pm0.00(\%)$ |
| MILPnet/2 | H. Seq. L=1 | $230.56 \pm 28.35$ | $37.7 \pm 39.8$ | $16.5 \pm 15.5$ | $0.21\pm 1.23(\%)$ | $146.0 \pm 26.6$ | $1.0 \pm 0.0$ | $1.7 \pm 0.2$ | $0.00\pm0.00(\%)$ |
| MILPnet/2 | H. Seq. L=2 | $230.56 \pm 28.35$ | $37.7 \pm 39.8$ | $16.5 \pm 15.6$ | $0.21\pm 1.23(\%)$ | $146.0 \pm 26.6$ | $1.0 \pm 0.0$ | $1.5 \pm 0.2$ | $0.00\pm0.00(\%)$ |
| MILPnet/2 | H. Seq. L=3 | $230.56 \pm 28.35$ | $38.5 \pm 41.0$ | $16.2 \pm 15.4$ | $0.17\pm 1.21(\%)$ | $146.0 \pm 26.6$ | $1.0 \pm 0.0$ | $1.5 \pm 0.2$ | $0.00\pm0.00(\%)$ |
| MILPnet/2 | H. Seq. L=4 | $230.56 \pm 28.35$ | $38.3 \pm 42.0$ | $16.3 \pm 15.5$ | $0.17\pm 1.21(\%)$ | $146.0 \pm 26.6$ | $1.0 \pm 0.0$ | $1.5 \pm 0.2$ | $0.00 \pm0.00(\%)$ |
| MILPnet/3 | H. Seq. L=1 | $230.56 \pm 28.35$ | $37.7 \pm 39.9$ | $16.5 \pm 15.5$ | $0.21\pm 1.23(\%)$ | $146.0 \pm 26.6$ | $1.0 \pm 0.0$ | $1.7 \pm 0.2$ | $0.00\pm0.00(\%)$ |
| MILPnet/4 | H. Seq. L=1 | $230.56 \pm 28.35$ | $37.7 \pm 39.8$ | $16.5 \pm 15.5$ | $0.21\pm 1.23(\%)$ | $146.0 \pm 26.6$ | $1.0 \pm 0.0$ | $1.7 \pm 0.2$ | $0.00\pm0.00(\%)$ |

**Very Large Heterogeneous solving generalization**
To assess heterogeneous generalization capability, we integrate simple *Heterogeneous MILPnet Variant* (Details in Appendix C.2) as a novel representation framework that replaces graph-based representations in advanced ML algorithms training, such as Con-PAS(Huang et al., 2024). Trained on the SC dataset by ConPAS training pipeline, MILPnet is directly transferred to solve very large MILP instances from open-source benchmarks (Gleixner et al., 2021) by predicting candidate solution, and refined the rounded solution by a quick heuristic for short time, then constructing the trust-region (Han et al., 2023)

Table 9: Heterogeneous generalization on *very-large benchmarks* (Gleixner et al., 2021) within 1500s with ConPAS style training.

| Method | 30n20b8 18.4K | | blp-ic98 13.6K | | blp-ar98 16.0K | |
|---|---|---|---|---|---|---|
| | Time | Obj | Time | Obj | Time | Obj |
| SCIP | 163.96 | 302.00 | 1500.15 | 4620.13 | 1500.10 | 6215.35 |
| ConPAS(GCN) | 175.86 | 302.00 | 1500.00 | 4817.66 | 1500.01 | 6254.08 |
| ConPAS(MILPnet) | **94.64** | **302.00** | **1500.00** | **4588.51** | **1500.01** | **6220.57** |

for final solving. The results, including primal bound iterations (Fig.11 in Appendix) and solving efficiency (Table 9), demonstrate that MILPnet generalizes stably and scalably across heterogeneous, large-scale benchmarks, achieving consistent performance improvements in all tested cases. Other Details are in Appendix C.2.

## 7 CONCLUSION

We propose *MILPnet*, a novel multi-scale hybrid framework for representing MILP problems through sequence modeling, rather than conventional graph methods. We prove that this architecture can approximate essential MILP mappings for arbitrary instances. Empirical evaluation confirms that *MILPnet* outperforms graph-based methods in terms of efficiency and performance, while addressing the Foldable MILPs where graph-based approaches fail.

ACKNOWLEDGEMENTS

The authors would like to thank the anonymous reviewers for their valuable feedback. This work was partially supported by the National Natural Science Foundation of China (Grant No. 62276024), the Beijing Natural Science Foundation (Grant No. 4262066), the Fundamental Research Funds for the Central Universities, Jilin University(Grant No. 93K172025K01), and the Fundamental Research Funds for the Central Universities (Grant No. 2025CX01010).

REPRODUCIBILITY STATEMENT

All datasets, experiments, and architecture hyperparameters used in our experiments are documented in the Appendix K and Appendix B.

ETHICS STATEMENT

This work is fundamental research in optimization and machine learning. All experiments were conducted without any potential risks. We do not foresee any direct negative societal impact or ethical concerns arising from this work.

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

# A  DETAILED ARCHITECTURE FOR MILPNET

Figure 5 illustrates the detailed architecture of MILPnet, which connects a feed-forward network with the core Multi-scale based Hybrid Attention.

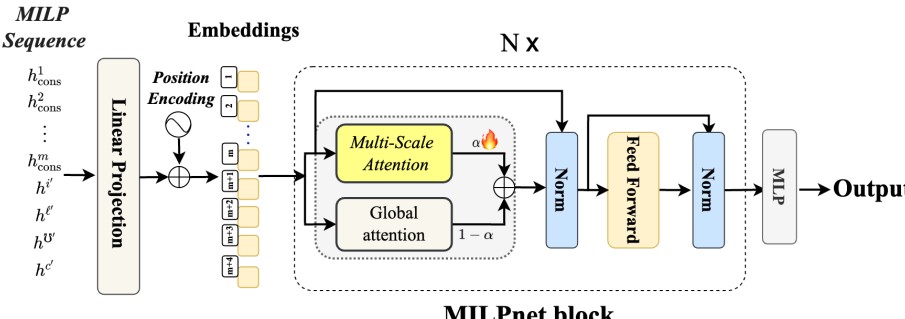

Figure 5: Detailed Architecture of *MILPnet*.

# B  FULL RESULTS ON MAXIMUM WINDOW SIZE, ATTENTION BLOCKS, REPRESENTATION, AND GENERALIZATION

## B.1  ABLATION STUDIES ON THE $\eta_{max}$

We conducted a study on the maximum sliding-window size and visualized the resulting representation curves in Figure 6.

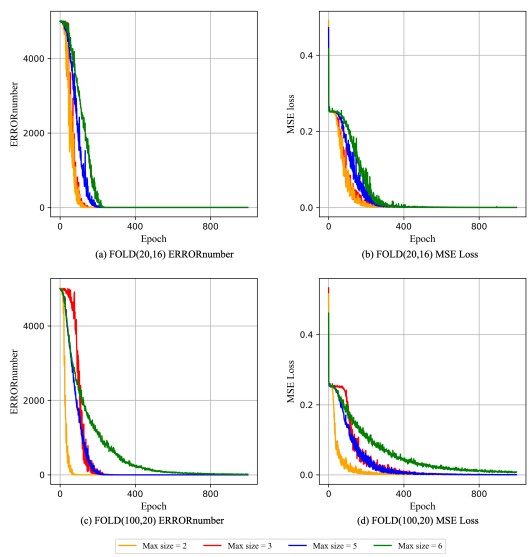

Figure 6: Performance of varied max window size of MILPnet on FOLD(20,16) and FOLD(100,20) for feasibility training. We can observe that the maximum sliding window size affects the convergence speed of the feasibility mapping approximation. The window size does not follow a simple pattern, but rather requires balancing.

## B.2 Performance on the number of attention blocks $L$

We conduct the ablation study on the number of attention blocks on FOLD(20) and FOLD(50).

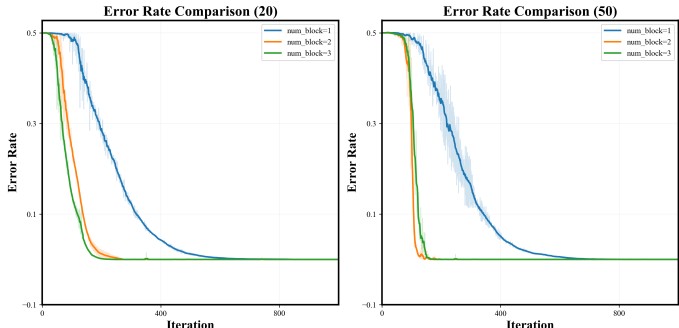

Figure 7: Performance of varied numbers of multi-scale attention blocks in MILPnet on FOLD(20,16) and FOLD(50,20) for feasibility with the same embedding size and same window size for blocks =1,2,3.

Table 10: Ablation study on the number of multi-scale attention blocks ($L$).

| Number of blocks | Method + Arch | FOLD(20,16) | | FOLD(50,20) | |
|---|---|---|---|---|---|
| | | MSE | ErrorN | MSE | ErrorN |
| GCN (Original) | GNN + Graph | 0.3070 | 5000 | 0.4719 | 5000 |
| L=1 | MILPnet + Seq | 0.0006 | 0 | 0.0003 | 0 |
| L=2 | MILPnet + Seq | 0.0001 | 0 | 0.0001 | 0 |
| L=3 | MILPnet + Seq | 0.0004 | 0 | 0.0001 | 0 |

## B.3 Permutation sensitivity on variable orders

Table 11: Experiments on variable permutation invariance. "V-Or *" represents randomly permutated variable order in the MILP-sequence."Original *"represents the original order.

| Order | Method + Arch | FOLD(20,16) | | FOLD(50,20) | |
|---|---|---|---|---|---|
| | | MSE | ErrorN | MSE | ErrorN |
| GCN (Original) | GNN + Graph | 0.3070 | 5000 | 0.4719 | 5000 |
| Original | MILPnet + Seq | 0.0003 | 0 | 0.0003 | 0 |
| V-Or 1 | MILPnet + Seq | 0.0006 | 0 | 0.0001 | 5 |
| V-Or 2 | MILPnet + Seq | 0.0003 | 0 | 0.0003 | 0 |
| V-Or 3 | MILPnet + Seq | 0.0004 | 0 | 0.0005 | 3 |

## B.4 Representation experiments

We conducted representation experiments on the representation of feasible mapping, the optimal solution mapping, and the objective optimal value, for MILP instances. The embedding sizes used in our experiments were chosen from $\{32, 128, 216, 256, 512\}$. Figure 8 shows the representation results of feasible mapping, Figure 9 shows the results of optimal solution mapping, and Figure 14 shows the results of the objective optimal value mapping. The dropout is used in this part.

## B.5 Generalization experiments results

We provide the full results on FOLD(20,) to FOLD(50,) in Table 8 following.

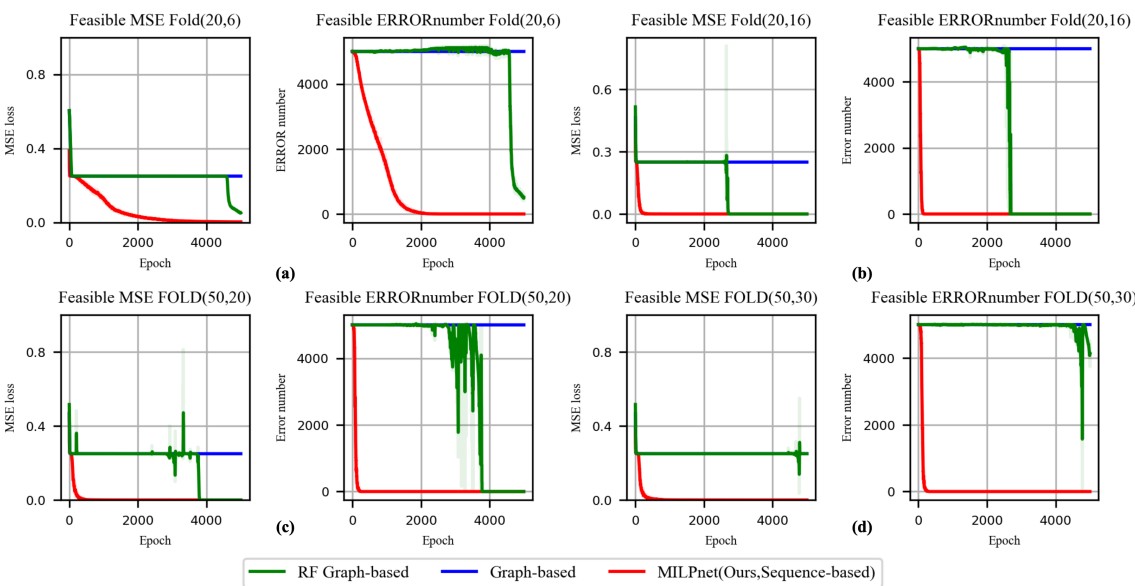

Figure 8: Representation experiments on feasibility for FOLD20 ((a) for FOLD(20,6), (b) for FOLD(20,16)) and FOLD50 ((c) for FOLD(50,20),(d) for FOLD(50,30)). **MILPnet** approximates the feasible mapping of Foldable MILP instances more efficiently.

Table 12: Generalization experiments for feasibility mapping on FOLD(20,*) to FOLD(50,*), with 10,000 foldable MILP instances from each.3M(20)/10M(50) means pre-train time for FOLD20 is 3mins, FOLD50 is 10mins;5M(20)/30M(50) means pre-train time for FOLD20 is 5mins, FOLD50 is 30mins.

| Method | Type | FOLD(20,6) | | | FOLD(20,16) | | | FOLD(50,20) | | | FOLD(50,30) | | |
|---|---|---|---|---|---|---|---|---|---|---|---|---|---|
| | | MSE | ErrorN | Params | MSE | ErrorN | Params | MSE | ErrorN | Params | MSE | ErrorN | Params |
| SCIP | Exact | —— | 0 | —— | —— | 0 | —— | —— | 0 | —— | —— | 0 | —— |
| GCN$_{3M(20)/10M(50)}$ | Graph | 0.3073 | 5000 | 1.21M | 0.3073 | 5000 | 1.21M | 0.3583 | 5000 | 1.21M | 0.3556 | 5000 | 1.21M |
| GCN$_{5M(20)/30M(50)}$ | Graph | 0.3073 | 5000 | 1.21M | 0.3070 | 5000 | 1.21M | 0.4719 | 5000 | 1.21M | 0.3242 | 5000 | 1.21M |
| GIN$_{3M(20)/10M(50)}$ | Graph | 0.4200 | 5000 | 1.63M | 0.4209 | 5000 | 1.63M | 0.4200 | 5000 | 1.63M | 0.4204 | 5000 | 1.63M |
| GIN$_{5M(20)/30M(50)}$ | Graph | 0.4199 | 5000 | 1.63M | 0.2999 | 5000 | 1.63M | 0.2939 | 5000 | 1.63M | 0.4204 | 5000 | 1.63M |
| SAGE$_{3M(20)/10M(50)}$ | Graph | 0.4642 | 5000 | 0.66M | 0.4999 | 5000 | 0.66M | 0.4714 | 5000 | 0.66M | 0.4586 | 5000 | 0.66M |
| SAGE$_{5M(20)/30M(50)}$ | Graph | 0.4642 | 5000 | 0.66M | 0.4999 | 5000 | 0.66M | 0.4714 | 5000 | 0.66M | 0.4856 | 5000 | 0.66M |
| PGN$_{3M(20)/10M(50)}$ | Graph | 0.2508 | 5000 | 1.64M | 0.2523 | 5000 | 1.64M | 0.2511 | 5000 | 1.64M | 0.2511 | 5000 | 1.64M |
| PGN$_{5M(20)/30M(50)}$ | Graph | 0.2511 | 5000 | 1.64M | 0.2511 | 5000 | 1.64M | 0.2512 | 5000 | 1.64M | 0.2512 | 5000 | 1.64M |
| GraphGPS$_{3M(20)/10M(50)}$ | Graph | 0.2500 | 5000 | 0.66M | 0.2500 | 5000 | 0.66M | 0.2500 | 5000 | 0.66M | 0.2500 | 5000 | 0.66M |
| GraphGPS$_{5M(20)/30M(50)}$ | Graph | 0.2500 | 5000 | 0.66M | 0.2500 | 5000 | 0.66M | 0.2500 | 5000 | 0.66M | 0.2500 | 5000 | 0.66M |
| GCN$^{rf}_{5M(20)/30M(50)}$ | Rf Graph | 0.2498 | 5000 | 1.21M | 0.2500 | 5000 | 1.21M | 0.2476 | 4334 | 1.21M | 0.5223 | 5000 | 1.21M |
| GCN$^{rf}_{5M(20)/30M(50)}$ | Rf Graph | 0.2126 | 2853 | 1.21M | 0.2499 | 4921 | 1.21M | 0.1177 | 0 | 1.21M | 0.2402 | 0 | 1.21M |
| GIN$^{rf}_{3M(20)/10M(50)}$ | Rf Graph | 0.2500 | 5000 | 1.63M | 0.2501 | 5000 | 1.63M | 0.2500 | 5000 | 1.63M | 0.4204 | 5000 | 1.63M |
| GIN$^{rf}_{5M(20)/30M(50)}$ | Rf Graph | 0.2499 | 4757 | 1.63M | 0.2500 | 5000 | 1.63M | 0.2500 | 5000 | 1.63M | 0.2458 | 2603 | 1.63M |
| SAGE$^{rf}_{3M(20)/10M(50)}$ | Rf Graph | 0.2499 | 5009 | 0.66M | 0.2499 | 4995 | 0.66M | 0.2500 | 4997 | 0.66M | 0.2500 | 5002 | 0.66M |
| SAGE$^{rf}_{5M(20)/30M(50)}$ | Rf Graph | 0.2499 | 5009 | 0.66M | 0.2499 | 4995 | 0.66M | 0.2500 | 4998 | 0.66M | 0.2500 | 4999 | 0.66M |
| PGN$^{rf}_{3M(20)/10M(50)}$ | Rf Graph | 0.2582 | 5000 | 1.64M | 0.2560 | 5000 | 1.64M | 0.2502 | 5000 | 1.64M | 0.2502 | 5000 | 1.64M |
| PGN$^{rf}_{5M(20)/30M(50)}$ | Rf Graph | 0.2507 | 5000 | 1.64M | 0.2514 | 5000 | 1.64M | 0.2512 | 5000 | 1.64M | 0.2502 | 5000 | 1.64M |
| GraphGPS$^{rf}_{3M(20)/10M(50)}$ | Rf Graph | 0.2510 | 5000 | 0.66M | 0.2510 | 5000 | 0.66M | 0.2500 | 5000 | 0.66M | 0.2502 | 5000 | 0.66M |
| GraphGPS$^{rf}_{5M(20)/30M(50)}$ | Rf Graph | 0.2510 | 5000 | 0.66M | 0.2510 | 5000 | 0.66M | 0.2500 | 5000 | 0.66M | 0.2520 | 5000 | 0.66M |
| MILPnet$_{3M(20)/10M(50)}$ | Ours (Seq) | 0.0005 | **0** | **0.56M** | 0.0004 | **0** | **0.56M** | 0.0005 | **0** | **0.60M** | 0.0023 | 12 | **0.60M** |
| MILPnet$_{5M(20)/30M(50)}$ | Ours (Seq) | **0.0003** | **0** | **0.56M** | **8.53e-5** | **0** | **0.56M** | **0.0005** | **0** | **0.60M** | 0.0082 | **0** | **0.60M** |

## B.6    CROSS-SIZE ADAPTION

Our approach also enables *End to End Cross-Size* generalization from FOLD(n,$m_1$) to FOLD(n,$m_2$), which is quite difficult for Bipartite Graph-based MILP *End-to-End* representation methods(Yehudai et al., 2021). We directly transfer the network pre-trained on FOLD (50,20) on the feasibility mapping to FOLD (50,30). As shown by the heatmap, the pre-trained network also easily achieves good performance on FOLD (50,30), which demonstrates that our model can still perform inference directly when faced with changes in length, effectively transferring its prior knowledge. This verifies that our network can effectively extract features; however, we also found that this adaptive capability is

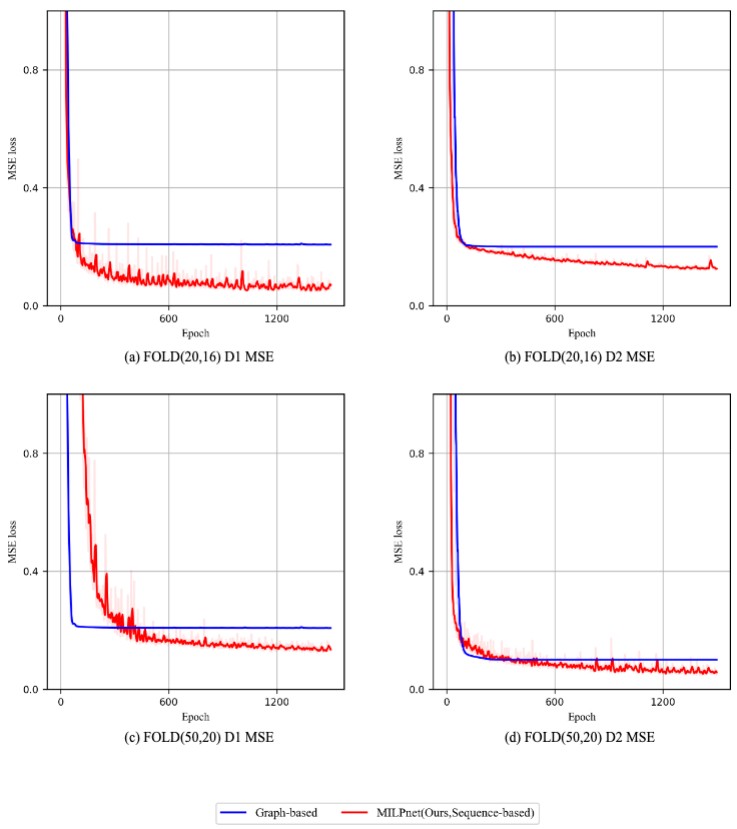

Figure 9: Representation experiments on optimal solution for FOLD20 and FOLD50 on D1 and D2 (details is shown in Appendix F). MILPnet approximates the solution mapping of Foldable MILP instances with smaller errors than graph-based method.

sensitive to factors such as model capacity and problem scale, which we plan to investigate further in future work.

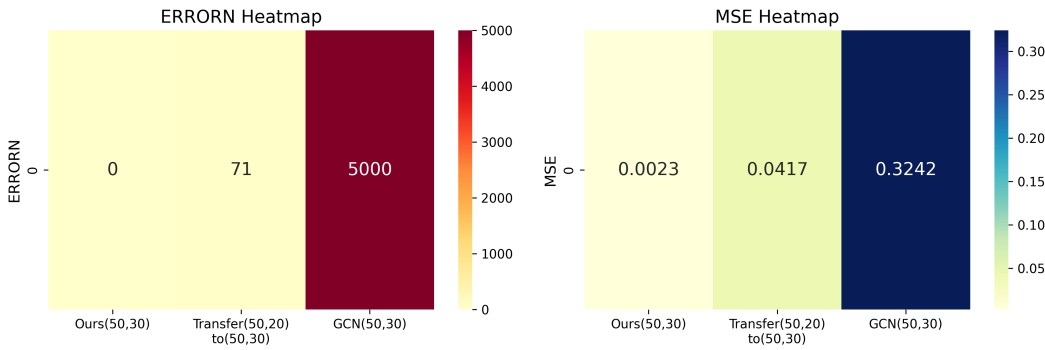

Figure 10: Cross-size adaptation for MILPnet.

## C  MILPNET SOLVING EFFICIENCY RESULTS

### C.1  ADDITIONAL EXPERIMENTS ON REAL-WORLD BENCHMARKS

This part, we use MILPnet and advanced graph-based network for predicting the near-optimal solution and refine by a local heuristic algorithm, the results are summarized in following Table 13. This result

Table 13: Additional evaluations with more powerful GNN baselines, including GIN, PGN, and GraphGPS, on the Real world practical problems

|  | IP | | SC | | CA | | FA | |
|---|---|---|---|---|---|---|---|---|
|  | Time | Gap | Time | Gap | Time | Gap | Time | Gap |
| GIN | 0.6254 | 0.0320 | 0.8223 | 0.6616 | 3.3065 | 0.7655 | 8.8688 | 0.7504 |
| PGN | 0.6023 | 0.1020 | 1.0492 | 0.4207 | 4.8369 | 1.0759 | 12.4009 | 0.7438 |
| GraphGps | 0.7324 | 0.0595 | 1.9202 | 0.4292 | 4.5980 | 1.0472 | 8.1973 | 0.8328 |
| **MILPnet**(Ours) | **0.1864** | **0.0234** | **0.6915** | **0.3483** | **3.2300** | **0.7651** | **1.3773** | **0.3503** |

demonstrate that MILPnet consistently outperforms these advanced graph-based models in terms of solving efficiency and inference speed. This further validates the effectiveness of the sequence-based architecture and the representation advantage of MILP-sequence. For SC and CA with 1000 variables and 500 constraints in ConPAS style training, the number of the dataset are: training on 10000 instances, solving on 50 instances.

### C.2  CONPAS STYLE HETEROGENEOUS VARIANT ON REAL-WORLD INSTANCES SOLVING

To address complex heterogeneous optimization problems in the real world, we train the heterogeneous *MILPnet* as the solution representation predictors on SC problems and construct the trust region solving on three very large benchmarks from (Gleixner et al., 2021). The visualization of the solving results in terms of primal bound trajectories is shown in Figure 11. Notably, because of the very-large scale of the instances, we use 5s pre-heuristic refinement based on GCN/MILPnet rounded prediction. The solving quality depends on the pretrained MILPnet predictions, which provide the trust-region. Since ConPAS's training code has not been released, we reproduce it ourselves. For ConPAS(MILPnet), we use the MSE loss based Contrastive loss for our training, which is related to our solution mappings approximation.

**Heterogeneous *MILPnet* Variant.** The heterogeneous *MILPnet* variant focuses on transfer capability. We observe that the dataset used for pre-training can limit this. Therefore, we relaxed the direction in the $n + 2$ dimension, so the MILP-sequence becomes $(m + 4) \times (n + 1)$ dimensional, to enhance its transfer capability. We use the same-scale Setcover to train this variant and use it as the solution predictor.

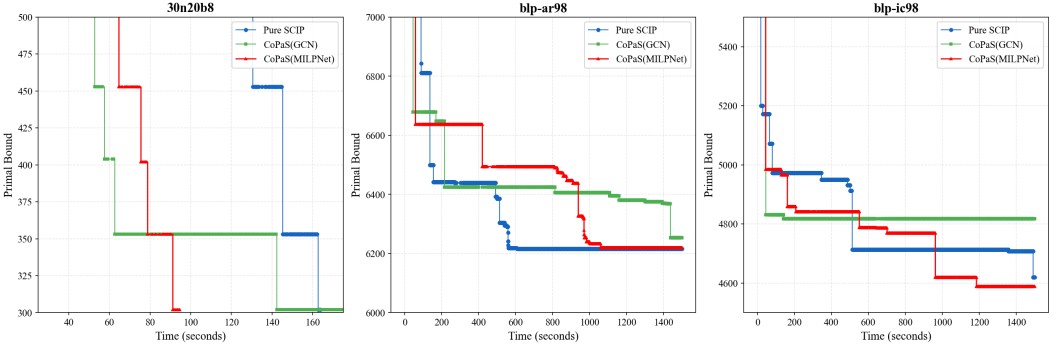

Figure 11: Primal bound iterations on very-large benchmarks solving with ConPAS(MILPnet)+ Trust Region +SCIP and ConPAS(GCN)+Trust Region + SCIP

# D GPU ANALYSIS AND OVERALL PERFORMANCE COMPARISON

## D.1 GPU MEMORY ANALYSIS

Figure 12 visualizes the GPU usage during the training process.

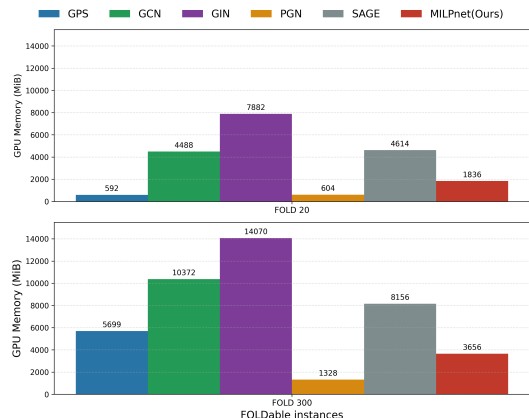

Figure 12: GPU memory usage on FOLD(20 ,*) to FOLD(300 ,*). **(The lower the better)**

## D.2 OVERALL PERFORMANCE COMPARISON

We conducted a comprehensive evaluation of various performance dimensions on the instances of FOLD300, and the results are shown in Figure 13.

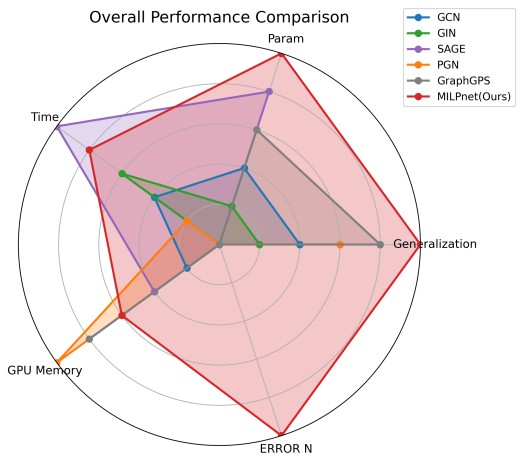

Figure 13: Overall performance comparisons on FOLD(300,*).**(The Broader the better)**

# E    PROOF OF THE SECTION 4.2 AND SECTION 4.3

We establish the results stated in Remark 4.2 and Section 4.3, where the former follows from Theorem 5 and the latter from Theorem 6 as follows.

**Theorem 5** (**Padding Equivalence**). *Define the topological space before padding as:* $H^{MILP0} = ((H^{cons})^m \cup H^{Var}) \times I \times H^{obj}$ *and the padded topological space as* $H^{MILP}$. $H^{MILP}$ *and* $H^{MILP0}$ *are homeomorphic (topologically equivalent):*

$$H^{MILP} \cong H^{MILP0} \tag{13}$$

*Proof.* Define $f$ as follows:

$$f((h_c, h_v, i, h_o)) = \left( h_c, \left( h_v, \underbrace{0, \ldots, 0}_{k \text{ times}} \right), \left( i, \underbrace{0, \ldots, 0}_{k \text{ times}} \right), \left( h_o, \underbrace{0, \ldots, 0}_{k \text{ times}} \right) \right),$$

where:

- $h_c \in (H^{cons})^m$,
- $h_v \in H^{Var}$,
- $i \in I$,
- $h_o \in H^{obj}$,
- $0, \ldots, 0 \in \{0\}^k$ are the padded zero dimensions.

Define $f^{-1}$ as:

$$f^{-1} \left( \left( h_c, \left( h_v, \underbrace{0, \ldots, 0}_{k \text{ times}} \right), \left( i, \underbrace{0, \ldots, 0}_{k \text{ times}} \right), \left( h_o, \underbrace{0, \ldots, 0}_{k \text{ times}} \right) \right) \right) = (h_c, h_v, i, h_o).$$

Subsequently, we introduce **(i)** and **(ii)**, which are instrumental in establishing the main theorem.

### (i) $f$ is Bijective

*Proof.* $f$ **is Injectivity:** Suppose $f(h_1) = f(h_2)$ for $h_1, h_2 \in H^{MILP0}$. Then,

$$f(h_1) = f(h_2) \implies (h_{c1}, (h_{v1}, 0, \ldots, 0), (i_1, 0, \ldots, 0), (h_{o1}, 0, \ldots, 0)) \tag{14}$$
$$= (h_{c2}, (h_{v2}, 0, \ldots, 0), (i_2, 0, \ldots, 0), (h_{o2}, 0, \ldots, 0)). \tag{15}$$

This equality implies:

$$h_{c1} = h_{c2}, \quad h_{v1} = h_{v2}, \quad i_1 = i_2, \quad h_{o1} = h_{o2}.$$

Therefore, $h_1 = h_2$, establishing injectivity.

$f$ **is Surjectivity:** For any $h' \in H^{MILP}$, suppose $h' = (h_c, (h'_v, p_v), (i', p_i), (h'_o, p_o))$, where $p_v, p_i, p_o \in \{0\}^k$. Then, there exists $h \in H^{MILP0}$ such that:

$$h = (h_c, h'_v, i', h'_o).$$

Applying $f$ to $h$, we obtain:

$$f(h) = (h_c, (h'_v, 0, \ldots, 0), (i', 0, \ldots, 0), (h'_o, 0, \ldots, 0)) = h'.$$

Thus, $f$ is surjective.

**(ii) $f$ and $f^{-1}$ are both continuous** *Proof.* Both $H^{MILP0}$ and $H^{MILP}$ are equipped with the product topology. In the product topology, a function is continuous if and only if each of its component functions is continuous.

**Continuity of $f$:** The mapping $f$ involves embedding each component of $H^{MILP0}$ into a higher-dimensional space by appending zero vectors. Each such embedding is continuous because it is defined by coordinate-wise inclusion and fixed assignments (adding zeros). Specifically:

$$f_i : H_i \to H_i \times \{0\}^k$$

is continuous for each component $H_i \in \{H^{\text{cons}}, H^{\text{Var}}, I, H^{\text{obj}}\}$.

**Continuity of $f^{-1}$:** The inverse mapping $f^{-1}$ involves projecting each padded component back to its original space by removing the appended zero vectors. Projection maps in the product topology are continuous. Specifically:

$$f_i^{-1} : H_i \times \{0\}^k \rightarrow H_i$$

is continuous for each component $H_i$.

Drawing upon the results presented in **(i)** and **(ii)**, the $f$ is bijective and both $f$ and $f^{-1}$ are continuous, $f$ is a homeomorphism. Therefore, the topological spaces $H^{\text{MILP}}$ and $H^{\text{MILP0}}$ are homeomorphic:

$$H^{\text{MILP}} \cong H^{\text{MILP0}}.$$

$\square$

**Theorem 6** (Constraint Permutation Invariance). *Let $\left(H_1^{cons}, H_2^{cons}, \ldots, H_m^{cons}, I, H^{Var}, H^{obj}\right)$ be topological spaces corresponding respectively to the* m *constraint feature spaces $H_i^{cons}$, the integer-index space $I'$, the variable-bounds space $H^{Var}$, and the objective-coefficient space $H^{obj}$. For any permutation $\sigma$ on the index set $\{1, 2, \ldots, m, 'I', 'Var', 'obj'\}$, their product spaces are homeomorphic:*

$$H_1^{cons} \times \cdots \times H_m^{cons} \times I' \times H^{Var'} \times H^{obj'} \cong \prod_{k \in \sigma(\{1,\ldots,m,I',Var',obj'\})} H^{(factor)_k}.$$

*In other words, the topological structure of the full MILP-sequence feature space is invariant under permutation of its component spaces.*

*Proof* The ordered product is

$$X = H_1^{\text{cons}} \times \cdots \times H_m^{\text{cons}} \times I' \times H^{\text{Var'}} \times H^{\text{obj'}},$$

Let

$$X_\sigma = \prod_{k \in \sigma(\{1,\ldots,m,I',Var',obj'\})} H^{(\text{factor})_k}$$

be an arbitrary permutation of the factors, where $\sigma$ is a permutation of the index set.

We explicitly construct a *coordinate-exchange (or coordinate-reordering) map*

$$\Phi : X \longrightarrow X_\sigma.$$

Given a point

$$x = \left(h_1^{\text{cons}}, \ldots, h_m^{\text{cons}}, i', h^{\text{Var'}}, h^{\text{obj'}}\right) \in X,$$

where

$$h_i^{\text{cons}} \in H_i^{\text{cons}}, \quad i' \in I, \quad h^{\text{Var}} \in H^{\text{Var'}}, \quad h^{\text{obj'}} \in H^{\text{obj'}},$$

define $\Phi(x)$ by rearranging these coordinate components according to $\sigma$. Concretely, if $\sigma$ sends the index 1 to position $\sigma(1)$, the index 2 to position $\sigma(2)$, etc., then

$$\Phi(x) = \big(\underbrace{h_{\text{the factor with index } \sigma^{-1}(1)}^{(\text{factor})}, \cdots\big)}_{\text{ordered according to } \sigma}.$$

In simpler terms, $\Phi$ reorders the factors $\left(h_1^{\text{cons}}, \ldots, h_m^{\text{cons}}, i, h^{\text{Var}}, h^{\text{obj}}\right)$ into the sequence $(\ldots)$ determined by $\sigma$.

**Bijection.** $\Phi$ is a *bijection* because:

- *Injective*: If $\Phi(x_1) = \Phi(x_2)$, then their coordinates in each position of the reordered product are identical. Since a product space comparison equates each factor, it follows that $x_1 = x_2$.

- *Surjective*: Given an arbitrary point $y \in X_\sigma$, we can "reverse reorder" its factors to form $x \in X$. By construction, $\Phi(x) = y$.

Because $\Phi$ is bijective and both $\Phi$ and $\Phi^{-1}$ are continuous, $\Phi$ is a *homeomorphism*. Hence

$$H_1^{\mathrm{cons}} \times \cdots \times H_m^{\mathrm{cons}} \times I' \times H^{\mathrm{Var'}} \times H^{\mathrm{obj'}} \; \cong \; \prod_{k \in \sigma(\{1,\ldots,m,\mathrm{I'},\mathrm{Var'},\mathrm{obj'}\})} H^{(\mathrm{factor})_k},$$

$\square$

**Theorem 7** (Variable Permutation Invariance)**.** *Let $H^{MILP}$ be the topological space of MILP-sequences defined in Section 4.3. For any permutation $\pi$ on the variable indices $\{1, 2, \ldots, n\}$, the MILP-sequence space is homeomorphic under variable permutations:*

$$H^{MILP} \; \cong \; H_\pi^{MILP},$$

*where $H_\pi^{MILP}$ denotes the space after permuting variable coordinates according to $\pi$. Consequently, the feasibility mapping $\Phi_{feas}$, optimal objective value mapping $\Phi_{obj}$, and optimal solution mapping $\Phi_{solu}$ are all invariant under variable permutation.*

*Proof.* We explicitly construct a *variable-coordinate reordering map*

$$\Psi \colon H^{\mathrm{MILP}} \; \longrightarrow \; H_\pi^{\mathrm{MILP}}.$$

Given an MILP-sequence

$$\mathbf{x} = [h_{\mathrm{cons}}^1, \ldots, h_{\mathrm{cons}}^m, h^{i'}, h^{\ell'}, h^{\mho'}, h^{c'}] \in H^{\mathrm{MILP}},$$

where each component is a vector in $\mathbb{R}^{n+2}$, we define $\Psi(\mathbf{x})$ by applying the coordinate permutation to each token:

For each token $h = (h_1, \ldots, h_n, h_{n+1}, h_{n+2}) \in \mathbb{R}^{n+2}$ in the MILP-sequence, we define the coordinate permutation operator $P_\pi \colon \mathbb{R}^{n+2} \to \mathbb{R}^{n+2}$ by

$$P_\pi(h) = (h_{\pi(1)}, \ldots, h_{\pi(n)}, h_{n+1}, h_{n+2}),$$

where $\pi$ permutes only the first $n$ variable-related coordinates, leaving the last two padding coordinates unchanged. This operator applies uniformly to all tokens: constraint tokens $h_i^{\mathrm{cons}}$, integer set token $h^{i'}$, variable bounds tokens $h^{\ell'}$ and $h^{\mho'}$, and objective token $h^{c'}$.

Then

$$\Psi(\mathbf{x}) = [P_\pi(h_{\mathrm{cons}}^1), \ldots, P_\pi(h_{\mathrm{cons}}^m), P_\pi(h^{i'}), P_\pi(h^{\ell'}), P_\pi(h^{\mho'}), P_\pi(h^{c'})] \in H_\pi^{\mathrm{MILP}}.$$

Then we prove that $\Psi$ is *bijective* and continuous:

1. **Bijection.** $\Psi$ is a *bijection* because:
   - *Injective*: If $\Psi(\mathbf{x}_1) = \Psi(\mathbf{x}_2)$, then for each token, the permuted coordinates are identical. Since $\pi$ is a bijection, the original coordinates must also be identical, hence $\mathbf{x}_1 = \mathbf{x}_2$.
   - *Surjective*: Given an arbitrary point $\mathbf{y} \in H_\pi^{\mathrm{MILP}}$, we can apply the inverse permutation $\pi^{-1}$ to each coordinate of each token to obtain $\mathbf{x} \in H^{\mathrm{MILP}}$ such that $\Psi(\mathbf{x}) = \mathbf{y}$.

2. **Continuity.** Both $\Psi$ and $\Psi^{-1}$ are continuous because coordinate permutation is a continuous operation in the product topology of $\mathbb{R}^{n+2}$. Specifically, for each token space $\mathbb{R}^{n+2}$, the permutation map is a linear isomorphism, and the composition over all tokens preserves continuity in the product space $H^{\mathrm{MILP}} \subset \mathbb{R}^{(m+4)(n+2)}$.

Therefore, $\Psi$ is a *homeomorphism*, and

$$H^{\mathrm{MILP}} \; \cong \; H_\pi^{\mathrm{MILP}}.$$

$\square$

**Remark E.1** (Permutation Invariance)**.** Theorems 6 and 7 establish that $H^{\mathrm{MILP}}$ is invariant under both *constraint permutation* (reordering tokens) and *variable permutation* (reordering coordinates within tokens). These homeomorphisms ensure that different representations of the same MILP instance are topologically equivalent and define identical optimization problems. Consequently, the mappings $\Phi_{\mathrm{feas}}$, $\Phi_{\mathrm{obj}}$, and $\Phi_{\mathrm{solu}}$ are invariant under both types of permutations, which is crucial for designing permutation-equivariant neural architectures.

# F    PROOF OF THE MAPPINGS.

We introduced the details of the MILP-sequence mapping definitions: with the defined topological space $H^{\text{MILP}}$, which encapsulates the MILP-sequence, we can define key mappings essential for analyzing MILP instances. These mappings assess feasibility, compute optimal objective values, and identify optimal solutions, providing a comprehensive toolkit for MILP analysis.

**Definition 8.** (Feasibility mapping by sequence) The feasibility mapping is a classification function that determines whether a sequence within $H^{\text{MILP}}$ represents a feasible solution:

$$\Phi_{\text{feas}} : H^{\text{MILP}} \to \{0, 1\} \tag{16}$$

where $\Phi_{\text{feas}} = 1$ indicates that the MILP instance is feasible.

**Definition 9.** (Optimal objective value mapping by sequence) The optimal objective value mapping for each MILP instance is defined as:

$$\Phi_{\text{obj}} : H^{\text{MILP}} \to \mathbb{R} \cup \{\infty, -\infty\} \tag{17}$$

which projects feasible sequences to their respective optimal values.

**Definition 10.** (Optimal solution mapping by sequence) To simplify the discussion, we focus on settings where all components of the vectors $\ell$ and $u$ are finite. This assumption ensures the existence of an optimal solution when the MILP problem is feasible. Consequently, we define a restricted subset of the MILP topological space, $\widetilde{H}^{\text{MILP}} \subset H^{\text{MILP}}$, which only has finite variable bounds. The optimal solution mapping for the MILP-sequence is defined as: For any $F \in \Phi_{\text{obj}}^{-1}(\mathbb{R})$, the MILP problem has a unique optimal solution with the smallest $\ell_2$-norm. Let

$$\Phi_{\text{solu}} : \widetilde{H}^{\text{MILP}} \cap \Phi_{\text{feas}}^{-1}(1) \to \mathbb{R}^n \tag{18}$$

Then we prove the feature mappings of the MILP-sequence are measurable in the following theorems.

**Theorem 11.** *The feasibility mapping for MILP-sequence is measurable.*

*Proof* The target space $\{0, 1\}$ is equipped with the discrete $\sigma$-algebra, where every subset is Borel. Specifically, the Borel sets in $\{0, 1\}$ are:

$$\mathcal{B}_{\{0,1\}} = \{\emptyset, \{0\}, \{1\}, \{0, 1\}\} .$$

We consider each possible Borel set $B \subseteq \{0, 1\}$ and examine $\Phi_{\text{feas}}^{-1}(B)$:

1. $B = \emptyset$:
$$\Phi_{\text{feas}}^{-1}(\emptyset) = \emptyset,$$
   which is trivially a Borel set.

2. $B = \{1\}$:

$$\Phi_{\text{feas}}^{-1}(\{1\}) = \left\{ (x, h) \in H^{\text{MILP}} \,\middle|\, \text{the MILP instance is feasible} \right\} .$$

   The feasibility of a MILP instance is determined by the existence of solutions that satisfy all linear constraints and integrality conditions. Specifically, it requires that there exists $x \in \mathbb{R}^n$ such that:
   $$Ax \leq b, \quad \ell \leq x \leq u, \quad x_j \in \mathbb{Z} \,\forall j \in \mathcal{I},$$
   where $\mathcal{I}$ is the set of indices corresponding to integer variables.

   The set of feasible MILP instances can be expressed as the intersection of:
   - A finite union of closed half-spaces defined by the linear constraints $Ax \leq b$.
   - Closed intervals defined by the variable bounds $\ell \leq x \leq u$.
   - Discrete conditions $x_j \in \mathbb{Z}$ for integer variables.

   Since finite unions and intersections of closed sets are closed (hence Borel), and the discrete conditions correspond to countable intersections, $\Phi_{\text{feas}}^{-1}(\{1\})$ is a Borel set in the product topology of $\mathbb{R}^{(m+4)(n+2)} \times H^{\text{MILP}}$.

3. $B = \{0\}$:

$$\Phi_{\text{feas}}^{-1}(\{0\}) = \left\{ (x, h) \in H^{\text{MILP}} \,\middle|\, \text{the MILP instance is infeasible} \right\}.$$

The infeasibility set is the complement of the feasibility set:

$$\Phi_{\text{feas}}^{-1}(\{0\}) = \left( \Phi_{\text{feas}}^{-1}(\{1\}) \right)^c.$$

Since $\Phi_{\text{feas}}^{-1}(\{1\})$ is a Borel set and the complement of a Borel set is also a Borel set, $\Phi_{\text{feas}}^{-1}(\{0\})$ is Borel.

4. $B = \{0, 1\}$:

$$\Phi_{\text{feas}}^{-1}(\{0, 1\}) = H^{\text{MILP}},$$

which is the entire space and hence a Borel set.

Since for every Borel set $B \subseteq \{0, 1\}$, the preimage $\Phi_{\text{feas}}^{-1}(B)$ is a Borel set in $\mathbb{R}^{(m+4)(n+2)} \times H^{\text{MILP}}$, the feasibility mapping $\Phi_{\text{feas}}$ is measurable. $\qquad\square$

**Remark F.1.** *The measurability of $\Phi_{feas}$ is crucial for ensuring that probabilistic and statistical analyses involving MILP instances are well-defined. Since $\Phi_{feas}$ maps measurable spaces to a discrete space with a simple $\sigma$-algebra, its measurability guarantees that feasibility can be reliably incorporated into broader measure-theoretic frameworks.*

**Remark F.2.** *Both the domain $H^{MILP}$ and the codomain $\{0, 1\}$ are equipped with their respective $\sigma$-algebras. The domain utilizes the product topology, and $H^{MILP}$ itself is a product of measurable spaces as defined earlier. The codomain $\{0, 1\}$ employs the discrete $\sigma$-algebra, where all subsets are measurable.*

**Theorem 12.** *The optimal objective value mapping for MILP-sequence is measurable.*

*Proof* To prove that the optimal objective value mapping $\Phi_{\text{obj}} : H^{\text{MILP}} \to \mathbb{R} \cup \{\infty, -\infty\}$ is measurable, we need to demonstrate that for every Borel set $B \subseteq \mathbb{R} \cup \{\infty, -\infty\}$, the preimage $\Phi_{\text{obj}}^{-1}(B)$ is a Borel set in $\mathbb{R}^{(m+4)(n+2)} \times H^{\text{MILP}}$.

The codomain $\mathbb{R} \cup \{\infty, -\infty\}$ can be equipped with the extended real line topology, where the Borel $\sigma$-algebra is generated by the open intervals in $\mathbb{R}$ along with the points $\{\infty\}$ and $\{-\infty\}$. The Borel sets in $\mathbb{R} \cup \{\infty, -\infty\}$ include:

1. $B \subseteq \mathbb{R}$
2. $B$ contains $\infty$ and/or $-\infty$

We consider each category of Borel sets in $\mathbb{R} \cup \{\infty, -\infty\}$ and examine $\Phi_{\text{obj}}^{-1}(B)$.

1. $B \subseteq \mathbb{R}$:

$$\Phi_{\text{obj}}^{-1}(B) = \left\{ (x, h) \in H^{\text{MILP}} \,\middle|\, \text{the optimal objective value of the MILP instance is in } B \right\}.$$

Assuming that the MILP's optimal objective value is determined by a continuous optimization process (which holds under certain regularity conditions, such as linearity of the objective function and constraints), $\Phi_{\text{obj}}$ can be considered a continuous function on the feasible set. Therefore, the preimage of any Borel set $B \subseteq \mathbb{R}$ under $\Phi_{\text{obj}}$ is a Borel set in the domain.

2. $B$ contains $\infty$ and/or $-\infty$:

$$\Phi_{\text{obj}}^{-1}(B) = \left\{ (x, h) \in H^{\text{MILP}} \,\middle|\, \Phi_{\text{obj}}(x, h) \in B \right\}.$$

The inclusion of $\infty$ or $-\infty$ typically corresponds to the infeasibility or unboundedness of the MILP instance:

- If $\Phi_{\text{obj}}(x, h) = \infty$, the MILP instance is unbounded above.
- If $\Phi_{\text{obj}}(x, h) = -\infty$, the MILP instance is unbounded below.

These conditions define specific subsets of the domain:

$$\Phi_{\text{obj}}^{-1}(\{\infty\}) = \left\{ (x,h) \in H^{\text{MILP}} \ \middle| \ \text{MILP is unbounded above} \right\},$$

and

$$\Phi_{\text{obj}}^{-1}(\{-\infty\}) = \left\{ (x,h) \in H^{\text{MILP}} \ \middle| \ \text{MILP is unbounded below} \right\}.$$

Assuming that the conditions for unboundedness are also defined by Borel sets (similar to feasibility), these preimages are Borel sets in the domain.

Since for every Borel set $B \subseteq \mathbb{R} \cup \{\infty, -\infty\}$, the preimage $\Phi_{\text{obj}}^{-1}(B)$ is a Borel set in $\mathbb{R}^{(m+4)(n+2)} \times H^{\text{MILP}}$, the mapping $\Phi_{\text{obj}}$ is measurable. $\qquad\square$

Before we give the proof of the mensurability of the optimal solution mapping for the MILP-sequence,we first introduce the Jankov-von Neumann Measurable Selection Theorem.

**Theorem 13** (**Jankov-von Neumann Measurable Selection Theorem**,(von Neumann, 1949)). *Let* $(X, \mathcal{A})$ *and* $(Y, \mathcal{B})$ *be measurable spaces, and let* $S : X \to 2^Y$ *be a measurable set-valued map such that for all* $x \in X$, $S(x)$ *is non-empty and closed in* $Y$. *Then, there exists a measurable function* $f : X \to Y$ *such that* $f(x) \in S(x)$ *for all* $x \in X$.

**Remark F.3.** *The **Jankov-von Neumann Measurable Selection Theorem** provides a crucial guarantee in measure theory and its applications. Given a measurable space $X$ and a set-valued mapping $A : X \to 2^Y$ where each $A(x)$ is a non-empty set, the theorem ensures the existence of a measurable function $f$ that selects an element from each $A(x)$ in a measurable manner. Specifically, for almost every $x \in X$, the function $f$ assigns a value $f(x)$ that belongs to the set $A(x)$. This result is particularly useful in areas such as optimization, probability theory, and economics, where selecting measurable choices from a set of feasible options is essential.*

Then we prove the measurblity of the optimal solution mapping for the MILP-sequence.

**Theorem 14.** *The optimal solution mapping for the MILP-sequence is measurable.*

*Proof.* This part proves the optimal solution mapping for the MILP-sequence is measurable. To prove that the optimal solution mapping

$$\Phi_{\text{solu}} : \widetilde{H}^{\text{MILP}} \cap \Phi_{\text{feas}}^{-1}(1) \to \mathbb{R}^n$$

is measurable, we need to demonstrate that for every Borel set $B \subseteq \mathbb{R}^n$, the preimage

$$\Phi_{\text{solu}}^{-1}(B)$$

is a Borel set in $H^{\text{MILP}}$.

Consider the mapping $\Phi_{\text{solu}}$ as a selection function that assigns to each feasible MILP instance its unique optimal solution with the smallest $\ell_2$-norm. Formally, for each

$$(x,h) \in \widetilde{H}^{\text{MILP}} \cap \Phi_{\text{feas}}^{-1}(1),$$

there exists at least one $x^* \in \mathbb{R}^n$ such that $x^*$ is an optimal solution. We aim to select a unique $x^*$ for each instance in a measurable manner. With Theorem 13, for each

$$(x,h) \in \widetilde{H}^{\text{MILP}} \cap \Phi_{\text{feas}}^{-1}(1),$$

the set of optimal solutions

$$S(x,h) = \{x^* \in \mathbb{R}^n \mid x^* \text{ is an optimal solution for } (x,h)\}$$

is non-empty and closed, then there exists a measurable selection function

$$\Phi_{\text{solu}} : \widetilde{H}^{\text{MILP}} \cap \Phi_{\text{feas}}^{-1}(1) \to \mathbb{R}^n$$

such that

$$\Phi_{\text{solu}}(x,h) \in S(x,h)$$

for all

$$(x,h) \in \widetilde{H}^{\text{MILP}} \cap \Phi_{\text{feas}}^{-1}(1).$$

Then, we prove the Non-emptiness and Cloasedness of $\widetilde{H}^{\text{MILP}} \cap \Phi_{\text{feas}}^{-1}(1)$, detailed as:

1. **Non-emptiness:** By definition,

$$\widetilde{H}^{\text{MILP}} \cap \Phi_{\text{feas}}^{-1}(1)$$

   consists of MILP instances that are feasible and have finite bounds, ensuring that an optimal solution exists. Therefore, $S(x, h)$ is non-empty for all

$$(x, h) \in \widetilde{H}^{\text{MILP}} \cap \Phi_{\text{feas}}^{-1}(1).$$

2. **Closedness:** The set of optimal solutions $S(x, h)$ is closed in $\mathbb{R}^n$. This is because optimal solutions to MILP problems, defined by linear constraints and objective functions, form closed sets under standard topologies.

Given that both the closedness and non-emptiness conditions are satisfied, Theorem 13 ensures the existence of a measurable selection function $\Phi_{\text{solu}}$. Since $\Phi_{\text{solu}}$ is a measurable selection function by the theorem, for any Borel set $B \subseteq \mathbb{R}^n$,

$$\Phi_{\text{solu}}^{-1}(B) = \{(x, h) \in H^{\text{MILP}} \mid \Phi_{\text{solu}}(x, h) \in B\}$$

is a Borel set in $H^{\text{MILP}}$. Therefore, the mapping $\Phi_{\text{solu}}$ is measurable. $\qquad\square$

*After establishing the measurability of the feasibility, optimal objective value, and optimal solution mappings, we define the corresponding measurable mapping sets for each of these mappings as follows:*

**Definition 15.** (Measurable Mapping Set for Feasibility Mapping) The **Feasibility Mapping Set** consists of all measurable functions

$$\Phi_{\text{feas}} : H^{\text{MILP}} \to \{0, 1\},$$

where $\Phi_{\text{feas}}(x, h) = 1$ indicates that the MILP instance defined by $(x, h)$ is feasible, and $\Phi_{\text{feas}}(x, h) = 0$ indicates infeasibility. Formally, the set is defined as:

$$\mathcal{F}_{\text{feas}}^{\text{MILP}} = \left\{ \Phi_{\text{feas}} : H^{\text{MILP}} \to \{0, 1\} \,\middle|\, \Phi_{\text{feas}} \text{ is measurable} \right\}.$$

**Definition 16** (Measurable Mapping Set for Optimal Objective Value Mapping)**.** The **Optimal Objective Value Mapping Set** comprises all measurable functions

$$\Phi_{\text{obj}} : H^{\text{MILP}} \to \mathbb{R} \cup \{\infty, -\infty\},$$

which assign to each MILP instance $(x, h)$ its optimal objective value. Specifically,

$$\Phi_{\text{obj}}(x, h) = \begin{cases} c^T x^* & \text{if the MILP instance is feasible and bounded,} \\ \infty & \text{if the MILP instance is unbounded above,} \\ -\infty & \text{if the MILP instance is unbounded below.} \end{cases}$$

Formally, the set is defined as:

$$\mathcal{F}_{\text{obj}}^{\text{MILP}} = \left\{ \Phi_{\text{obj}} : H^{\text{MILP}} \to \mathbb{R} \cup \{\infty, -\infty\} \,\middle|\, \Phi_{\text{obj}} \text{ is measurable} \right\}.$$

**Definition 17** (Measurable Mapping Set for Optimal Solution Mapping)**.** The **Optimal Solution Mapping Set** consists of all measurable functions

$$\Phi_{\text{solu}} : \widetilde{H}^{\text{MILP}} \cap \Phi_{\text{feas}}^{-1}(1) \to \mathbb{R}^n,$$

which assign to each feasible and bounded MILP instance $(x, h)$ its unique optimal solution $x^*$ with the smallest $\ell_2$-norm. Formally, the set is defined as:

$$\mathcal{F}_{\text{solu}}^{\text{MILP}} = \left\{ \Phi_{\text{solu}} : \widetilde{H}^{\text{MILP}} \cap \Phi_{\text{feas}}^{-1}(1) \to \mathbb{R}^n \,\middle|\, \Phi_{\text{solu}} \text{ is measurable} \right\}.$$

## G    THE PROOF OF THE SECTION 5.2.

Firstly, theorem Lusin is presented before the proof in this section.

**Theorem 18** (Lusin's Theorem,(Aliprantis & Border, 2006)). *Let $f : \mathbb{R} \to \mathbb{R}$ be a measurable function on a measurable set $E \subset \mathbb{R}$ with finite measure. For every $\epsilon > 0$, there exists a closed set $C \subset E$ such that the measure of $E \setminus C$ is less than $\epsilon$, and $f$ restricted to $C$ is continuous.*

Secondly, for the probability measure used in Theorems 2–4 we would like give the Remark G.1 to clarify the probability using:

**Remark G.1** (On the Probability Measure). The probability $P(\cdot)$ in Theorems 2–4 accounts for the randomness in neural network training, including random initialization and stochastic optimization (SGD). Specifically, $P$ is the probability measure over the space of trained networks induced by the random training process. This formulation aligns with the Probably Approximately Correct learning. For a fixed trained network $F_{\text{HYA}}$, the probability can be interpreted as the empirical error rate on the finite dataset $D$.

Then we give the proof of the theorems in the Section 5.2.

### G.1    PROOF AND COROLLARY ON THEOREM 2

***Proof of Theorem 2*** We leverage the measurability of the feasibility mapping $\Phi_{\text{feas}}$, the Jankov-von Neumann Measurable Selection Theorem, and Lusin's Theorem to construct a neural network $F_{\text{HYA}}$ that approximates $\Phi_{\text{feas}}$ with high accuracy on the finite dataset $D$.

From our earlier definitions in Definition 15, $\Phi_{\text{feas}} : H^{\text{MILP}} \to \{0, 1\}$ is measurable. This ensures that $\Phi_{\text{feas}}$ is compatible with measure-theoretic frameworks. Lusin's Theorem states that for any measurable function and any $\delta > 0$, there exists a compact subset where the function is continuous and the measure of the complement is less than $\delta$. However, since $D$ is a finite dataset, we can consider the discrete measure where each point in $D$ has an equal probability mass.

Given the finiteness of $D$, Lusin's Theorem trivially holds as we can define $\delta = \epsilon$ and select the entire dataset $D$ as the compact subset where $\Phi_{\text{feas}}$ is continuous (since all functions on finite sets are continuous).

By the Universal Approximation Theorem (Hornik et al., 1989), Since our MILPnet is an architecture that combines at least one-layer feedforward network structure with activation functions, it can approximate any continuous function on a compact subset to arbitrary accuracy. Since $D$ is finite, and $\Phi_{\text{feas}}$ is effectively continuous on $D$, there exists a neural network $F_{\text{HYA}} \in \mathcal{F}_{\text{HYA}}^{\text{MILPnet}}$ that satisfies:

$$|F_{\text{HYA}}(x) - \Phi_{\text{feas}}(x)| < \frac{1}{2}, \quad \forall x \in D.$$

This ensures that:

$$\mathbb{I}_{F_{\text{HYA}}}(x) = \begin{cases} 1 & \text{if } F_{\text{HYA}}(x) > \frac{1}{2}, \\ 0 & \text{otherwise,} \end{cases}$$

matches $\Phi_{\text{feas}}(x)$ exactly for all $x \in D$. Since $D$ is finite, the probability $P$ can be interpreted as a uniform distribution over $D$. Given that $F_{\text{HYA}}$ correctly classifies all $x \in D$, we have:

$$P\left(\mathbb{I}_{F_{\text{HYA}}(x) > \frac{1}{2}} \neq \Phi_{\text{feas}}(x)\right) = 0 < \epsilon.$$

Thus, the constructed neural network $F_{\text{HYA}}$ satisfies the required condition for the theorem.    $\square$

**Remark G.2.** *While the above steps suffice for a finite dataset, the framework can be extended using the Jankov-von Neumann Measurable Selection Theorem for more general settings. This theorem ensures the existence of a measurable selection function that can be approximated by neural networks even in infinite-dimensional spaces, provided the feasibility mapping satisfies the necessary measurability and closedness conditions. However, for the scope of this theorem with a finite dataset $D$, the construction above is sufficient to guarantee the existence of the desired neural network $F_{\text{HYA}}$.*

*Then we have corollary on the infinite dataset for feasibility mapping:*

**Corollary 1** (Extension to infinite dataset). Let $D \subset H^{\text{MILP}}$ be an infinite or continuous dataset with a finite measure $\mu(D) < \infty$. For any $\epsilon > 0$, there exists a neural network $F_{\text{HYA}} \in \mathcal{F}_{\text{HYA}}^{\text{MILPnet}}$ such that:

$$P\left(\mathbb{I}_{F_{\text{HYA}}(x) > \frac{1}{2}} \neq \Phi_{\text{feas}}(x)\right) < \epsilon, \quad \forall x \in D, \tag{19}$$

where $x$ is the MILP-sequence, and $\Phi_{\text{feas}}(x)$ is the feasibility mapping of the MILP instance.

*Proof.* We have proven that the feasibility mapping $\Phi_{\text{feas}} : H^{\text{MILP}} \to \{0, 1\}$ is assumed to be measurable on $D$ in Theorem B.1.

By **Lusin's Theorem**, for the measurable function $\Phi_{\text{feas}}$ and for any $\epsilon > 0$, there exists a compact (closed and bounded) subset $C \subset D$ such that:

$$\mu(D \setminus C) < \epsilon$$

and $\Phi_{\text{feas}}$ restricted to $C$ is continuous:

$$\Phi_{\text{feas}}|_C : C \to \{0, 1\} \text{ is continuous.}$$

Since $\Phi_{\text{feas}}$ is a simple indicator function, its continuity on $C$ implies that $C$ avoids the boundary cases where the feasibility of the MILP instances changes.

Given that $\Phi_{\text{feas}}$ is continuous on $C$, by the Universal Approximation Theorem, there exists a neural network $F_{\text{HYA}} \in \mathcal{F}_{\text{HYA}}^{\text{MILPnet}}$ that approximates the indicator function $\mathbb{I}_{\Phi_{\text{feas}}(x)}$ to within an error less than $\frac{1}{2}$ on $C$. Specifically:

$$\left| F_{\text{HYA}}(x) - \mathbb{I}_{\Phi_{\text{feas}}(x)} \right| < \frac{1}{2}, \quad \forall x \in C$$

This implies that:

$$\mathbb{I}_{F_{\text{HYA}}(x) > \frac{1}{2}} \iff \mathbb{I}_{\Phi_{\text{feas}}(x)} = 1, \quad \forall x \in C$$

Thus, the neural network $F_{\text{HYA}}$ correctly classifies the feasibility of MILP instances in $C$. Consider the probability that the classification error occurs:

$$P\left(\mathbb{I}_{F_{\text{HYA}}(x) > \frac{1}{2}} \neq \Phi_{\text{feas}}(x)\right)$$

This event can only occur if $x \in D \setminus C$, since for $x \in C$, the classification is guaranteed to be correct. Therefore:

$$P\left(\mathbb{I}_{F_{\text{HYA}}}(x) > \frac{1}{2} \neq \Phi_{\text{feas}}(x)\right) \leq \mu(D \setminus C) < \epsilon$$

Thus, the probability that $F_{\text{HYA}}$ misclassifies any $x \in D$ is bounded by $\epsilon$. By constructing $F_{\text{HYA}}$ using Lusin's Theorem to ensure continuity on a large compact subset $C$ of $D$, and then applying the Universal Approximation Theorem to approximate the feasibility indicator function on $C$, we have established the existence of a neural network within $\mathcal{F}_{\text{HYA}}^{\text{MILPnet}}$ that satisfies the desired probabilistic bound for the classification task on infinite or continuous datasets $D$. $\qquad \square$

### G.2 Proof and corollary of Theorem 3

***Proof of Theorem 3*** To prove the theorem, we leverage the **Lusin's Theorem** alongside the Universal Approximation Theorem for neural networks. The proof is divided into two parts corresponding to the classification and regression problems.

**1. Classification Problem:**

We aim to construct a neural network $F_{\text{HYA},1}$ that accurately classifies whether the objective value $\Phi_{\text{obj}}(x)$ is finite for all $x \in D$.

First, we have proven that the feature mapping $\Phi_{\text{obj}} : H^{\text{MILP}} \to \mathbb{R} \cup \{\infty\}$ is measurable. By **Lusin's Theorem**, for any measurable function $f$ defined on a measurable set with finite measure, and for any $\epsilon > 0$, there exists a closed set $C \subset D$ such that:

$$\mu(D \setminus C) < \epsilon$$

and $f$ restricted to $C$ is continuous. Since $D$ is finite, the measure $\mu(D)$ is finite, and thus Lusin's Theorem is applicable. Therefore, there exists a subset $C \subset D$ where $\Phi_{\text{obj}}$ is continuous.

Given that $\Phi_{\text{obj}}$ is continuous on $C$, by the Universal Approximation Theorem, there exists a neural network $F_{\text{HYA},1} \in \mathcal{F}_{\text{HYA}}^{\text{MILPnet}}$ that approximates the indicator function $\mathbb{I}_{\Phi_{\text{obj}}(x) \in \mathbb{R}}$ to within an error less than $\epsilon$ on $C$. Specifically:

$$\left| F_{\text{HYA},1}(x) - \mathbb{I}_{\Phi_{\text{obj}}(x) \in \mathbb{R}} \right| < \frac{1}{2}, \quad \forall x \in C$$

Since $\mu(D \setminus C) < \epsilon$, the probability that $F_{\text{HYA},1}$ misclassifies any $x \in D$ is less than $\epsilon$:

$$P\left( \mathbb{I}_{F_{\text{HYA},1}(x) > \frac{1}{2}} \neq \mathbb{I}_{\Phi_{\text{obj}}(x) \in \mathbb{R}} \right) < \epsilon, \quad \forall x \in D$$

**2. Regression problems:**

Next, we construct a neural network $F_{\text{HYA},2}$ to predict the objective value $\Phi_{\text{obj}}(x)$ with an error less than $\delta$ for all $x \in D$ where $\Phi_{\text{obj}}(x)$ is finite.

We have proven that the $\Phi_{\text{obj}}$ is measurable in Theorem B.2. By **Lusin's Theorem**, for the regression task, there exists a closed subset $C' \subset D \cap \Phi_{\text{obj}}^{-1}(\mathbb{R})$ such that:

$$\mu\left( (D \cap \Phi_{\text{obj}}^{-1}(\mathbb{R})) \setminus C' \right) < \epsilon$$

and $\Phi_{\text{obj}}$ is continuous on $C'$.

By the Universal Approximation Theorem, there exists a neural network $F_{\text{HYA},2} \in \mathcal{F}_{\text{HYA}}^{\text{MILPnet}}$ that approximates $\Phi_{\text{obj}}$ to within an error $\delta$ on $C'$:

$$|F_{\text{HYA},2}(x) - \Phi_{\text{obj}}(x)| < \delta, \quad \forall x \in C'$$

Since the measure of the complement set is less than $\epsilon$, the probability that the prediction error exceeds $\delta$ is bounded by $\epsilon$:

$$P\left( |F_{\text{HYA},2}(x) - \Phi_{\text{obj}}(x)| > \delta \right) < \epsilon, \quad \forall x \in D \cap \Phi_{\text{obj}}^{-1}(\mathbb{R})$$

By constructing $F_{\text{HYA},1}$ and $F_{\text{HYA},2}$ using Lusin's Theorem to ensure measurability and continuity on large subsets, and then applying the Universal Approximation Theorem to approximate the respective functions, we have established the existence of neural networks within $\mathcal{F}_{\text{HYA}}^{\text{MILPnet}}$ that satisfy the desired probabilistic bounds for both classification and regression tasks on the finite dataset $D$. $\quad\square$

*Then we give the corollary on the infinite dataset for optimal objective mapping as follows:*

**Corollary 2** (Extension to Compact Infinite Sets). Let $K \subset H^{\text{MILP}}$ be a compact subset equipped with the measure $\mu_{H^{\text{MILP}}}$ defined in Section 4.2. Define the probability measure $P$ on $K$ by normalizing: $P(A) = \frac{\mu_{H^{\text{MILP}}}(A)}{\mu_{H^{\text{MILP}}}(K)}$ for any measurable set $A \subset K$.

For any $\epsilon, \delta > 0$, there exist two neural networks $F_{\text{HYA},1}, F_{\text{HYA},2} \in \mathcal{F}_{\text{HYA}}^{\text{MILPnet}}$ such that:

**1. Classification of finite objective values:**

$$P\left( \mathbb{I}_{F_{\text{HYA},1}(x) > \frac{1}{2}} \neq \mathbb{I}_{\Phi_{\text{obj}}(x) \in \mathbb{R}} \right) < \epsilon, \quad \forall x \in K \tag{20}$$

**2. Regression of objective values:**

$$P\left( |F_{\text{HYA},2}(x) - \Phi_{\text{obj}}(x)| > \delta \right) < \epsilon, \quad \forall x \in K \cap \Phi_{\text{obj}}^{-1}(\mathbb{R}) \tag{21}$$

*Proof.* The proof parallels Theorem 3, replacing the counting measure on finite $D$ with the probability measure $P$ on compact $K$.

**1. Classification Problem.**

Since $\Phi_{\text{obj}} : H^{\text{MILP}} \to \mathbb{R} \cup \{\infty\}$ is measurable and $K$ is compact with $\mu_{H^{\text{MILP}}}(K) < \infty$, we apply Lusin's Theorem.

For any $\epsilon > 0$, there exists a closed set $C \subset K$ such that:

$$P(K \setminus C) = \frac{\mu_{H^{\text{MILP}}}(K \setminus C)}{\mu_{H^{\text{MILP}}}(K)} < \frac{\epsilon}{2}$$

and $\Phi_{\text{obj}}|_C$ is continuous. By the Universal Approximation Theorem, there exists $F_{\text{HYA},1} \in \mathcal{F}_{\text{HYA}}^{\text{MILPnet}}$ such that:

$$\sup_{x \in C} \left| F_{\text{HYA},1}(x) - \mathbb{I}_{\Phi_{\text{obj}}(x) \in \mathbb{R}} \right| < \frac{1}{4}$$

This ensures that for all $x \in C$:

$$\mathbb{I}_{F_{\text{HYA},1}(x) > \frac{1}{2}} = \mathbb{I}_{\Phi_{\text{obj}}(x) \in \mathbb{R}}$$

The set of misclassified points is contained in $K \setminus C$, therefore:

$$P\left( \mathbb{I}_{F_{\text{HYA},1}(x) > \frac{1}{2}} \neq \mathbb{I}_{\Phi_{\text{obj}}(x) \in \mathbb{R}} \right) \leq P(K \setminus C) < \epsilon$$

**2. Regression Problem.** Let $K_{\text{finite}} = K \cap \Phi_{\text{obj}}^{-1}(\mathbb{R})$. Define the conditional probability measure on $K_{\text{finite}}$ by:

$$P_{\text{finite}}(A) = \frac{\mu_{H^{\text{MILP}}}(A)}{\mu_{H^{\text{MILP}}}(K_{\text{finite}})}, \quad A \subset K_{\text{finite}}$$

By Lusin's Theorem applied to $\Phi_{\text{obj}}$ on $K_{\text{finite}}$, for any $\epsilon > 0$, there exists a closed set $C' \subset K_{\text{finite}}$ such that:

$$P_{\text{finite}}(K_{\text{finite}} \setminus C') < \frac{\epsilon}{2}$$

and $\Phi_{\text{obj}}|_{C'}$ is continuous. By the Universal Approximation Theorem, there exists $F_{\text{HYA},2} \in \mathcal{F}_{\text{HYA}}^{\text{MILPnet}}$ such that:

$$\sup_{x \in C'} |F_{\text{HYA},2}(x) - \Phi_{\text{obj}}(x)| < \delta$$

Therefore:

$$P_{\text{finite}}\left( |F_{\text{HYA},2}(x) - \Phi_{\text{obj}}(x)| > \delta \right) \leq P_{\text{finite}}(K_{\text{finite}} \setminus C') < \epsilon$$

$\square$

### G.3 Proof and the corollary of Theorem 4

***Proof of Theorem 4*** Since $D$ is finite, let us denote it as:

$$D = \{x_1, x_2, \ldots, x_n\}$$

for some integer $n \geq 1$. For each $x_i \in D$, $\Phi_{\text{solu}}(x_i)$ is a well-defined finite solution in $\mathbb{R}^n$.

The Universal Approximation Theorem states that a feedforward neural network with at least one hidden layer and a sufficient number of neurons can approximate any continuous function on compact subsets of $\mathbb{R}^n$ to any desired degree of accuracy, provided the activation function is non-linear (e.g., Sigmoid, ReLU).

Given that $D$ is finite, it is trivially compact. Therefore, there exists a neural network $F_{\text{HYA},W}$ that can approximate the mapping $\Phi_{\text{solu}}$ on $D$ with arbitrary precision. Specifically, for each $x_i \in D$, we can ensure:

$$\|F_{\text{HYA},W}(x_i) - \Phi_{\text{solu}}(x_i)\| < \delta$$

by appropriately choosing the network architecture and weights $W$.

Since $D$ is finite, the probability $P$ can be interpreted over a uniform distribution or any probability measure defined on $D$. However, because we have constructed $F_{\text{HYA},W}$ such that the approximation error is less than $\delta$ for every $x \in D$, the event

$$\|F_{\text{HYA},W}(x) - \Phi_{\text{solu}}(x)\| > \delta$$

does not occur for any $x \in D$. Therefore:

$$P\left( \|F_{\text{HYA},W}(x) - \Phi_{\text{solu}}(x)\| > \delta \right) = 0 < \epsilon$$

for any $\epsilon > 0$.

By the Universal Approximation Theorem, we can construct a neural network $F_{\text{HYA},W}$ that approximates the solution mapping $\Phi_{\text{solu}}$ on the finite dataset $D$ with an error less than $\delta$ for all $x \in D$. Consequently, the probability that the approximation error exceeds $\delta$ is zero, which is trivially less than any $\epsilon > 0$. This establishes the existence of such a neural network within $\mathcal{F}_{\text{HYA},V}^{\text{MILPnet}}$. $\square$

*Then we have the corollary on the infinite dataset:*

**Corollary 3** (Extension to infinite dataset). Let $D \subset \Phi_{\text{obj}}^{-1}(\mathbb{R}) \subset H^{\text{MILP}}$ be an infinite dataset with a finite measure $\mu(D) < \infty$. For any $\epsilon, \delta > 0$, there exists a neural network $F_{\text{HYA},W} \in \mathcal{F}_{\text{HYA},W}^{\text{MILPnet}}$ such that:

$$P\left(\|F_{\text{HYA},W}(x) - \Phi_{\text{solu}}(x)\| > \delta\right) < \epsilon, \quad \forall x \in D, \tag{22}$$

where $P$ denotes the probability measure on $D$.

*Proof.* To extend the theorem to infinite datasets, we employ **Lusin's Theorem** in conjunction with the Universal Approximation Theorem.

Assume $D$ is equipped with a probability measure $\mu$ such that $\mu(D) = 1$ (without loss of generality, as we can normalize the measure). Also, we have proven that the solution mapping $\Phi_{\text{solu}} : H^{\text{MILP}} \to \mathbb{R}^n$ is measurable on $D$ before.

By Lusin's Theorem, for the measurable function $\Phi_{\text{solu}}$ and for any $\epsilon > 0$, there exists a compact (closed and bounded) subset $C \subset D$ such that:

$$\mu(D \setminus C) < \epsilon$$

and $\Phi_{\text{solu}}$ restricted to $C$ is continuous:

$$\Phi_{\text{solu}}|_C : C \to \mathbb{R}^m \text{ is continuous.}$$

Since $C$ is compact and $\Phi_{\text{solu}}|_C$ is continuous, the Universal Approximation Theorem ensures that there exists a neural network $F_{\text{HYA},W} \in \mathcal{F}_{\text{HYA},W}^{\text{MILPnet}}$ such that:

$$\sup_{x \in C} \|F_{\text{HYA},W}(x) - \Phi_{\text{solu}}(x)\| < \delta.$$

This implies that for all $x \in C$:

$$\|F_{\text{HYA},W}(x) - \Phi_{\text{solu}}(x)\| < \delta.$$

Consider the probability that the approximation error exceeds $\delta$:

$$P\left(\|F_{\text{HYA},W}(x) - \Phi_{\text{solu}}(x)\| > \delta\right).$$

This event can only occur if $x \in D \setminus C$, since for $x \in C$, the error is guaranteed to be less than $\delta$. Therefore:

$$P\left(\|F_{\text{HYA},W}(x) - \Phi_{\text{solu}}(x)\| > \delta\right) \leq \mu(D \setminus C) < \epsilon.$$

Thus, we have:

$$P\left(\|F_{\text{HYA},W}(x) - \Phi_{\text{solu}}(x)\| > \delta\right) < \epsilon.$$

By Lusin's Theorem, we ensure that $\Phi_{\text{solu}}$ is continuous on a large subset $C$ of $D$. The Universal Approximation Theorem then guarantees the existence of a neural network $F_{\text{HYA},W}$ that approximates $\Phi_{\text{solu}}$ within $\delta$ on $C$. Consequently, the probability that the approximation error exceeds $\delta$ on the entire dataset $D$ is bounded by $\epsilon$.

This establishes that for infinite or continuous datasets with finite measure, there exists a neural network within $\mathcal{F}_{\text{HYA},V}^{\text{MILPnet}}$ that satisfies the desired probabilistic bound on the approximation error. $\square$

## H   PROOF OF THE STABILITY

In this section, we establish the stability properties of the MILP mappings $\Phi_{\text{feas}}$, $\Phi_{\text{obj}}$, and $\Phi_{\text{solu}}$ under small perturbations of problem coefficients. Our analysis builds upon Berge's Maximum Theorem (Aliprantis & Border, 2006).

**Theorem 19** (Berge's Maximum Theorem Aliprantis & Border (2006)). *Let $X$ and $T$ be topological spaces, and let $f : X \times T \to \mathbb{R}$ be continuous. Let $C : T \rightrightarrows X$ be a correspondence (set-valued map) such that:*

1. *$C(t)$ is non-empty and compact for all $t \in T$,*

2. *$C$ is continuous, i.e., its graph $Gr(C) = \{(t, x) : x \in C(t)\}$ is closed and $C$ is lower hemicontinuous.*

*Define the value function and argmax correspondence by:*

$$V(t) = \max_{x \in C(t)} f(x,t), \quad X^*(t) = \arg \max_{x \in C(t)} f(x,t) = \{x \in C(t) : f(x,t) = V(t)\}.$$

*Then:*

1. *$V(t)$ is continuous in $t$.*

2. *$X^*(t)$ is non-empty, compact-valued, and upper hemicontinuous in $t$.*

We first analyze the stability of $\Phi_{\text{obj}}$, which exhibits the strongest stability properties among the three mappings.

**Definition 20** (Continuous Relaxation for MILP sequence). For a MILP sequence $\mathbf{x} = [h_{\text{cons}}^1, \ldots, h_{\text{cons}}^m, h^{i'}, h^{\ell'}, h^{\mho'}, h^{c'}] \in H^{\text{MILP}}$, its *continuous relaxation* is the linear program obtained by dropping the integrality constraints:

$$\text{CR}(\mathbf{x}) : \quad \min_{x \in \mathbb{R}^n} \{\langle c, x \rangle : Ax \le b, \ \ell \le x \le u\},$$

where $A, b, c, \ell, u$ are extracted from the tokens in $\mathbf{x}$.

**Proposition 21** (Stability of Continuous Relaxation). Let $\mathbf{x} \in H^{\text{MILP}}$ be a feasible MILP instance whose continuous relaxation has a locally non-empty and uniformly bounded feasible region in a neighborhood of $\mathbf{x}$. Assume further that the feasible-region correspondence is continuous at $\mathbf{x}$. Then the optimal value function of the continuous relaxation, denoted $\Phi_{\text{obj}}^{\text{CR}}$, is continuous at $\mathbf{x}$. Furthermore, if the optimal solution is non-degenerate, then $\Phi_{\text{obj}}^{\text{CR}}$ is locally Lipschitz continuous with some constant $L_{\text{CR}} > 0$.

*Proof.* The continuous relaxation defines a parametric linear program where the parameter $t = (A, b, c, \ell, u)$ lives in $H^{\text{MILP}}$. Let $X = \mathbb{R}^n$ and $T = H^{\text{MILP}}$. The objective function $f(x,t) = \langle c, x \rangle$ is continuous in $(x,t)$. Then the feasible region correspondence $C(t) = \{x \in \mathbb{R}^n : Ax \le b, \ell \le x \le u\}$ satisfies:

- **Compactness:** By assumption, $C(t)$ is locally non-empty and uniformly bounded in a neighborhood of $\mathbf{x}$. Combined with the closed constraints, $C(t)$ is compact for all $t$ in this neighborhood.

- **Continuity:** By assumption, $C$ is continuous at $\mathbf{x}$; in particular, its graph $\text{Gr}(C)$ is closed and $C$ is both upper and lower hemicontinuous at $\mathbf{x}$.

By Berge's Theorem 19, $\Phi_{\text{obj}}^{\text{CR}}(\mathbf{x})$ is continuous at $\mathbf{x}$. Under non-degeneracy, the optimal basis remains unchanged in a neighborhood of $\mathbf{x}$. The optimal value is given by the basis solution $x^* = B^{-1}b_B$, where $B$ is the optimal basis matrix. The objective value is:

$$\Phi_{\text{obj}}^{\text{CR}}(\mathbf{x}) = \langle c_B, B^{-1}b_B \rangle,$$

which is a smooth function of the local problem coefficients within the neighborhood where the basis is constant.

For a perturbation $\mathbf{x}' = \mathbf{x} + \boldsymbol{\epsilon}$ with $\|\boldsymbol{\epsilon}\|_2 \le \delta$:

$$
\begin{aligned}
|\Phi_{\text{obj}}^{\text{CR}}(\mathbf{x}') - \Phi_{\text{obj}}^{\text{CR}}(\mathbf{x})| &= |\langle c', x'^* \rangle - \langle c, x^* \rangle| \\
&\le |\langle c' - c, x'^* \rangle| + |\langle c, x'^* - x^* \rangle| \\
&\le \|c' - c\|_2 \cdot \|x'^*\|_2 + \|c\|_2 \cdot \|x'^* - x^*\|_2 \\
&\le L_{\text{CR}} \cdot \delta,
\end{aligned}
$$

where $L_{\text{CR}} > 0$ is a local constant that absorbs the uniform bound on the optimal solutions and the local sensitivity of the unchanged optimal basis. $\square$

## H.1 STABILITY OF THE OPTIMAL OBJECTIVE MAPPING

**Theorem 22** (Conditional Stability of MILP Objective Value). *Let* $\mathbf{x} \in H^{MILP}$ *be a feasible MILP sequence with bounded feasible region. Consider a perturbation* $\mathbf{x}' = \mathbf{x} + \boldsymbol{\epsilon}$ *where* $\|\boldsymbol{\epsilon}\|_2 \leq \delta$. *If the following assumptions hold:*

1. **Feasibility preservation:** $\Phi_{feas}(\mathbf{x}') = 1$,

2. **Optimal solution stability:** *The optimal integer solution* $x^* \in \mathbb{Z}^n$ *of* $\mathbf{x}$ *remains feasible for* $\mathbf{x}'$,

3. **Common boundedness:** *the optimal solutions* $x^*$ *and* $x'^*$ *satisfy* $\|x'^*\|_2 \leq D$ *and* $\|x'^* - x^*\|_2 \leq D$,

*then:*

$$|\Phi_{obj}(\mathbf{x}') - \Phi_{obj}(\mathbf{x})| \leq (\|c\|_2 + \delta) \cdot D.$$

*Proof.* Let $x^* \in \mathbb{Z}^n$ be an optimal solution to $\mathbf{x}$, and $x'^* \in \mathbb{Z}^n$ be an optimal solution to $\mathbf{x}'$.

By condition (2), $x^*$ is feasible for $\mathbf{x}'$, so:

$$\Phi_{\text{obj}}(\mathbf{x}') = \langle c', x'^* \rangle \leq \langle c', x^* \rangle.$$

Similarly, $x'^*$ is feasible for its problem, and by condition (1), both $x^*$ and $x'^*$ lie in the bounded integer feasible region. Thus:

$$\Phi_{\text{obj}}(\mathbf{x}) = \langle c, x^* \rangle.$$

Now:

$$
\begin{aligned}
|\Phi_{\text{obj}}(\mathbf{x}') - \Phi_{\text{obj}}(\mathbf{x})| &= |\langle c', x'^* \rangle - \langle c, x^* \rangle| \\
&\leq |\langle c', x'^* \rangle - \langle c, x'^* \rangle| + |\langle c, x'^* \rangle - \langle c, x^* \rangle| \\
&= |\langle c' - c, x'^* \rangle| + |\langle c, x'^* - x^* \rangle| \\
&\leq \|c' - c\|_2 \cdot \|x'^*\|_2 + \|c\|_2 \cdot \|x'^* - x^*\|_2 \\
&\leq \delta \cdot D + \|c\|_2 \cdot D \\
&= (\|c\|_2 + \delta) \cdot D,
\end{aligned}
$$

where we used $\|c' - c\|_2 \leq \delta$ from the perturbation bound and $\|x'^*\|_2, \|x'^* - x^*\|_2 \leq D$ from condition 3. $\square$

## H.2 STABILITY OF THE FEASIBILITY MAPPING

We now analyze the stability of $\Phi_{\text{feas}}$.

**Theorem 23** (Conditional Stability of Feasibility). *Let* $\mathbf{x} \in H^{MILP}$ *with* $\Phi_{feas}(\mathbf{x}) = 1$. *Suppose the feasible integer points satisfy a* strict feasibility *condition: there exists* $\rho > 0$ *such that for all* $x^* \in Feas(\mathbf{x}) \cap \mathbb{Z}^n$,

$$Ax^* \leq b - \rho\mathbf{1}, \quad \ell + \rho\mathbf{1} \leq x^* \leq u - \rho\mathbf{1},$$

*where* $\mathbf{1}$ *is the all-ones vector. Then for any perturbation* $\mathbf{x}' = \mathbf{x} + \boldsymbol{\epsilon}$ *with* $\|\boldsymbol{\epsilon}\|_\infty \leq \delta < \min\left\{\rho, \frac{\rho}{M+1}\right\}$, *we have:*

$$\Phi_{feas}(\mathbf{x}') = 1.$$

*where* $M = \sup_{x^* \in Feas(\mathbf{x}) \cap \mathbb{Z}^n} \|x^*\|_1 < \infty$.

*Proof.* Let $x^* \in \text{Feas}(\mathbf{x}) \cap \mathbb{Z}^n$ be any feasible integer point for $\mathbf{x}$.

For the perturbed instance $\mathbf{x}'$ with parameters $(A', b', \ell', u')$, we have:

$$\|A' - A\|_\infty \leq \delta, \quad |b'_i - b_i| \leq \delta, \quad |\ell'_j - \ell_j| \leq \delta, \quad |u'_j - u_j| \leq \delta.$$

Check constraint satisfaction:

$$
\begin{aligned}
A'x^* = (A + \Delta A)x^* &= Ax^* + \Delta A x^* \\
&\leq (b - \rho\mathbf{1}) + \|\Delta A\|_\infty \|x^*\|_1 \mathbf{1} \\
&\leq b - \rho\mathbf{1} + \delta\|x^*\|_1 \mathbf{1}.
\end{aligned}
$$

Since $b' \geq b - \delta\mathbf{1}$ and $\|x^*\|_1 \leq M$, the condition $\delta(M+1) < \rho$ implies $A'x^* \leq b'$.

For sufficiently small $\delta < \rho$, we can ensure all integer points remain strictly feasible. Therefore, $\Phi_{\text{feas}}(\mathbf{x}') = 1$. $\qquad\square$

### H.3 STABILITY OF THE OPTIMAL SOLUTION MAPPING

Finally, we analyze $\Phi_{\text{solu}}$.

**Theorem 24** (Conditional Stability of Solution Mapping). *Let $\mathbf{x} \in H^{MILP}$ with optimal solution $x^* \in \mathbb{Z}^n$. Suppose $x^*$ is the* unique optimal solution *and satisfies a* strong optimality gap *condition: for all $x \in Feas(\mathbf{x}) \cap \mathbb{Z}^n$ with $x \neq x^*$:*

$$
\langle c, x \rangle > \langle c, x^* \rangle + \gamma,
$$

*for some $\gamma > 0$. Assume that the perturbation preserves $x^*$ and introduces no new feasible integer points. Then there exists $\delta > 0$ such that for perturbation $\mathbf{x}' = \mathbf{x} + \epsilon$ with $\|\epsilon\|_2 \leq \delta$:*

$$
\Phi_{solu}(\mathbf{x}') = x^*.
$$

*That is, the optimal solution remains unchanged.*

*Proof.* By assumption, the gap condition applies to any $x \in \text{Feas}(\mathbf{x}') \cap \mathbb{Z}^n$. For the perturbed instance with objective $c' = c + \Delta c$ where $\|\Delta c\|_2 \leq \delta$:

For any $x \in \text{Feas}(\mathbf{x}') \cap \mathbb{Z}^n$ with $x \neq x^*$:

$$
\begin{aligned}
\langle c', x \rangle - \langle c', x^* \rangle = \langle c + \Delta c, x \rangle - \langle c + \Delta c, x^* \rangle \\
= (\langle c, x \rangle - \langle c, x^* \rangle) + \langle \Delta c, x - x^* \rangle \\
\geq \gamma - \|\Delta c\|_2 \|x - x^*\|_2 \\
\geq \gamma - \delta \cdot D,
\end{aligned}
$$

where $D = \max_{x \in \text{Feas}(\mathbf{x}) \cap \mathbb{Z}^n,\, x \neq x^*} \|x - x^*\|_2$ is finite because the integer feasible set is bounded. If $\delta < \frac{\gamma}{D}$, then $\langle c', x \rangle > \langle c', x^* \rangle$ for all $x \neq x^*$, ensuring $x^*$ remains optimal for $\mathbf{x}'$. $\qquad\square$

## I DETAILS OF THE TIME-COMPLEXITY OF MILPNET

Excluding the linear transformation, the time complexity for multi-scale operations across all windows can be simplified to $O\left(d(m+4)\sum_{k=1}^{N} h \cdot \eta_k\right)$. The global attention mechanism employs a global multi-head self-attention mechanism over the entire sequence, resulting in a final time complexity of $O\left((m+4)\sum_{k=1}^{N} h\eta_k d + h(m+4)^2 d\right)$ for the hybrid attention. Given that the maximum window size satisfies $\eta_{\max} \leq m+4$, it follows that $\sum_{k=1}^{N} \eta_k \leq N \cdot (m+4)$. Therefore, the time complexity can be further simplified to $O\left(h \cdot (m+4)^2 d\,(N+1)\right)$, where $d$ represents the linear embedding size and $h$ denotes the number of attention heads.

## J SPARSE VARIANT

To mitigate the quadratic overhead of global attention, we evaluated a multi-scale sliding sparse attention (with sliding mask) to replace the global attention, and discovered that it can achieve faster inference while maintaining comparable performance. As the sparse sliding attention uses sparse masks to restrict each position's attention only to specific locations beyond the step interval $s$, the time complexity of the stride attention is:

$$
\mathcal{T}_{\text{stride}} = \mathcal{O}\left(\frac{h(m+4)^2 d}{s}\right) \tag{23}
$$

Table 14: Experimental results on FOLD(200,20) with 1-hour pre-training and a stride of 2

|  | Fold(200,20) | | |
|---|---|---|---|
|  | Error | Inference Time | MSE |
| GCN | 5000 | 0.228 | 0.268 |
| MILPnet(Sparse) | 763 | **0.033** | 0.091 |
| **MILPnet(ours)** | **191** | 0.117 | **0.016** |

Thus, the time complexity of multi-scale sliding spare attention becomes:

$$\mathcal{T}_{\text{sparse}} = \mathcal{O}\big(dh\big((m+4)\sum_{k=1}^{N}\eta_k + (m+4)^2/s\big)\big) \tag{24}$$

We can calculate the speed-up ratio against our original design as:

$$\text{Speedup} = \frac{\sum_{k=1}^{N}\eta_k + (m+4)}{\sum_{k=1}^{N}\eta_k + (m+4)/s} \tag{25}$$

In practice, $(m+4) \gg \sum_{k=1}^{N}\eta_k$ (sequence length is much larger than window size, in large scale MILPs), so the speed-up of the time-complexity is finally as:

$$\text{Speedup} \approx \frac{(m+4)}{(m+4)/s} = \boxed{s}$$

This signifies that the model's scalability with engineering optimizations.

## K DETAILS OF THE EXPERIMENTS

### K.1 DETAILED EXPERIMENT SETTINGS

**Baselines. As methods using variant GNNs for MILP representation are still limited**, we adapted several representative graph algorithms to serve as baselines. To validate the advantages of our sequence-based algorithm, we compare our model against multiple graph-based representation methods that **model MILPs as bipartite graphs**. The **GNN-based networks** include GCN (Chen et al., 2023b), GIN (Xu et al., 2019), and SAGE (Wu et al., 2021). The attention-based graph networks include PGN (Cappart et al., 2022) and GraphGPS (Wang et al., 2023b). We also include the **random feature graph modeling method** proposed in (Chen et al., 2023b), which is specifically designed to alleviate the feasibility prediction problem on Foldable instances.

**Metrics.** We established several evaluation metrics for the experimental settings of the various feature mappings discussed in the context of Mixed-Integer Linear Programming (MILP). (1)**Approximation Error**: For all feature mappings, we assessed the average approximation error, defined as the mean of prediction errors across all instances. (2) **Feasibility Error Number**: Specifically for feasibility mappings, we designed metrics to measure the error prediction rate and the number of prediction errors. (3)**Model Params**: We evaluated the size of the model parameters to assess the impact of different feature mappings on the model's complexity.

**Implementation.** The MILPnet module is implemented using PyTorch. Our experiments were conducted on a single NVIDIA 4090Ti GPU (24GB) and a 12th Gen Intel(R) Core(TM) i5-12600KF 3.69GHz CPU for Foldable instance experiments, for real-world MILP experiments, we use NVIDIA A800 GPU.

### K.2 DETAILS OF GENERATING FOLDABLE MILP INSTANCES

We strictly follow the well-designed foldable dataset generation method and the publicly available code implementation provided by Chen et al. (2023b), adhering to the formal definition of foldable MILP (Definition 4.1) therein. In details, we set the $c_1 = \cdots = c_n = 0$ foldable instances as D1, the $c_1 = \cdots = c_n = 0.01$ foldable instances as D2. We use D1 and D2 as our experimental datasets. It is worth mentioning that in the representation and generalization experiments on foldable instances, our training set contains 10,000 foldable instances, while the test set also includes 10,000 foldable

instances, with each set containing 5,000 feasible instances,every instance is generated with different LP files (i.e. different integer indices and different lb and ub).

**Variable Generation**: The lower and upper bounds for each variable $x_j$ are generated from a normal distribution. If the lower bound is greater than the upper bound, they are swapped. Some variables are specified as integer variables (i.e., $x_j \in \{0, 1\}$), while others are continuous variables, with lower and upper bounds. The continuous and integer variables, construct different MILP variables.

**Feasible and Infeasible Problem Constraint Setup**: We strictly follow publicly available code implementation provided by Chen et al. (2023b) for feasible and infeasible problem construction. For example, for the feasible problem ($k1$), the constraints are set as follows:

$$x_{j_1} + x_{j_2} = 1, \quad x_{j_2} + x_{j_3} = 1, \quad \ldots, \quad x_{j_n} + x_{j_1} = 1$$

These constraints connect binary variables with equality, forming a cyclic structure that guarantees the problem is feasible; For the infeasible problem ($k2$), the constraints are adjusted to make the problem infeasible. For example, in publicly available code implementation provided by Chen et al. (2023b), the basic infeasible construction logic from (Chen et al., 2023b) is:

$$x_{j_1} + x_{j_2} = 1, \quad x_{j_2} + x_{j_3} = 1, \quad x_{j_3} + x_{j_1} = 1$$

This setup results in an infeasible problem, which is basic but strong construction.

### K.3 DETAILS OF THE REPRESENTATION EXPERIMENTS

In the representation experiments of our MILPnet on Foldable instances. The graph-based methods share the same set of experimental seeds and details as MILPnet. And our feasibility dataset consists of 50% feasible instances and 50% infeasible instances, while other datasets are obtained by removing infeasible samples.

### K.4 DETAILS OF THE GENERALIZATION EXPERIMENTS

In the generalization experiments of our MILPnet on Foldable instances, the network is firstly pre-trained for on same-scale MILPs in training set. For the feasibility generalization experiments, we limited the pre-training time, setting it to 3 minutes and 5 minutes on FOLD20, 10 minutes and 30 minutes on FOLD50, and 1 hours on FOLD100 to FOLD500. We then compared the performance of our model with other baselines under different pre-training time constraints.

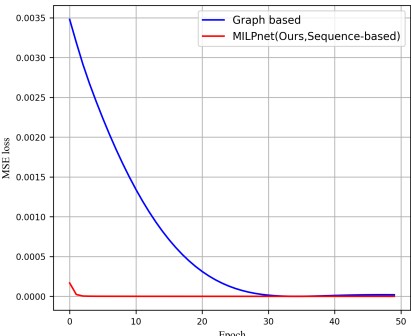

Figure 14: Representation experiments on the optimal objective value for FOLD20. MILPnet approximates the optimal value mapping of Foldable MILP instances with smaller errors than graph-based method.

### K.5 FOUR COMMON REAL-WORLD MILP SOLVING BENCHMARKS

This section introduces the details of constructing specific MILP instances. In particular, for the SC, CA, and FC problems, we follow the instance construction method described in Learn2Branch(Gasse et al., 2019), and the numbers of variables and constraints are shown in the table below.

Table 15: Number of variables and constraints for real-world benchmarks

| Dataset | Variables | Constraints |
|---------|-----------|-------------|
| IP | 1,083 | 195 |
| SC | 200 | 40 |
| CA | 300 | 40 |
| FC | 200 | 60 |

## L    ADDITIONAL UNFOLDABLE EXPERIMENTS.

Although this paper primarily conducts experiments on foldable MILP instances, we also perform representation experiments on unfoldable instances. Similarly, we compare our method with graph-based networks. In this part, the max window size is chosen as 2, and the MILPnet embedding size is chosen as 32, the graph-based methods (We select GCN) embedding size is 8.

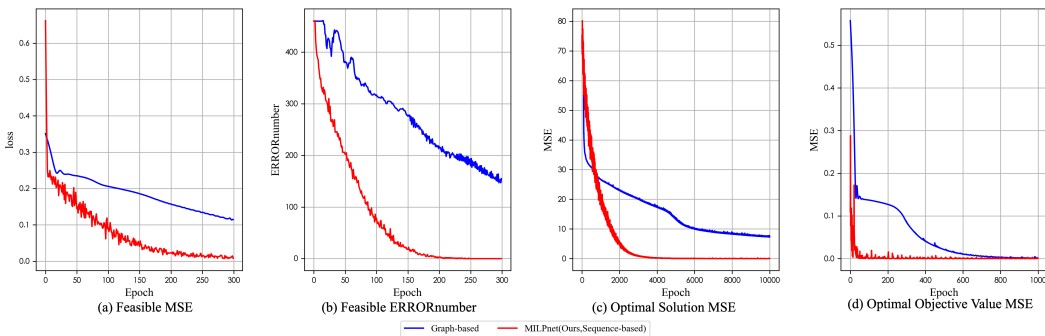

Figure 15: In the representation experiments on unfold20, MILPnet still outperforms graph-based methods with a smaller approximation error. Additionally, it maintains a smaller parameter size while achieving an estimation error lower than that of the graph-based method.

