# OpenReview forum: "MILPnet: A Multi-Scale Architecture with Geometric Feature Sequence Representations for Advancing MILP Problems"
_ICLR.cc/2026/Conference — ICLR 2026 Poster_

### Official Review · Reviewer_DtRt · 2025-10-30

**Soundness:** 4
**Presentation:** 4
**Contribution:** 4
**Rating:** 6
**Confidence:** 4

**Summary:**

The paper proposes MILPnet, a multi-scale hybrid-attention model that represents MILP instances as sequences of geometric features (constraints/objective), rather than graphs. The motivation is that message-passing GNNs are limited by 1-WL expressivity and can fail on families of “Foldable MILPs” that are indistinguishable under WL-type symmetries. MILPnet builds multi-scale views of these sequences and applies cross-scale attention to capture local and global structure. The authors prove universal approximation properties (feasibility, optimal value, solution mapping) on a measurable topological space and report large empirical gains over graph-based methods on Foldable MILPs, with fewer parameters and better convergence; they also show generalization across scales and integration into a solver pipeline.

**Strengths:**

1. Compelling motivation: explicit WL-type failure mode for GNNs on MILPs; sequence modeling avoids that bottleneck.
2. Multi-scale design: principled way to capture local/global geometric structure in constraint/objective sequences.
3. Theoretical guarantees beyond typical empirical claims.
4. Practical angle: improved convergence and smaller models; integration into solver workflows.

**Weaknesses:**

1. The comparison is not comprehensive. Please compare to modern ML-augmented B&B (e.g., branching/cuts/node-selection learned policies) and clarify where MILPnet plugs into the B&B stack (pre-solve heuristic? branching score? feasibility classifier?).
2. While the paper compares against standard GNN-based MILP representation methods, to convincingly establish state-of-the-art performance, it should also include or discuss recent baselines such as RL-based MILP solvers (e.g., RL-MILP Solver 2024), partial-assignment reduction frameworks (e.g., ConPaS 2023/24), solver-configuration learning methods (e.g., Separator-Learning 2023), and model-reduction methods (e.g., Learning Model Reduction 2024). Including these would strengthen the empirical claim of the universality of your sequence-based representation.
3. More ablation studies are required. Provide ablations on (i) feature choices per token, (ii) number and granularity of scales, (iii) attention depth vs. performance, and (iv) effect on different MILP families (set-covering, capacitated facility location, routing, scheduling).
4. Generalization tests are needed to demonstrate the practical potential for the work. Beyond Foldable MILPs, evaluate on large real-world sets and report time-to-first-feasible and optimality gap closed vs. SCIP/CPLEX.

**Questions:**

1. Where exactly in a production solver would you place MILPnet (branching score predictor, primal heuristic, node ranking, cut selection)? Please quantify wall-clock speedups and gap closure.
2. Is MILPnet permutation-invariant to constraint/variable orderings? If not, what data augmentation or canonicalization is used?
3. Can the theory be tightened to show sample complexity or stability under small perturbations of coefficients?
4. How sensitive is performance to scale hyper-parameters (window sizes, strides) and to sequence length (very large models)?
5. Could you hybridize with recent high-capacity GNNs (e.g., higher-order/WL-beyond) to reap graph-topology benefits while preserving sequence expressivity?

---

> ### Author Response · Authors · 2025-11-27
>
> We sincerely thank the reviewer for the valuable feedback. All revisions have been highlighted in blue in the updated manuscript.
>
> ---
>
> ## Response to Weakness 1 and Question 1
>
> Thank you for raising this important point. In our implementation, MILPnet is integrated into the branch-and-bound (B&B) workflow as a **machine learning-based pre-solve heuristic** in a neural diving style[1]. MILPnet warm-starts B&B by proposing high-quality partial assignments that define tighter trust regions, thereby reducing the search space before branching begins.
>
> We evaluated MILPnet on two larger public benchmarks and measured four key metrics: (i) number of B&B nodes explored, (ii) solving time, (iii) dual gap, and (iv) final objective value.
>
> The table below is the revised Table 9, where "/w" denotes the maximum scale size of MILPnet and "L" denotes the depth of the multi-scale attention mechanism.
>
> **Table 9: Branch and Bound performance of MILPnet (Neural Diving based pre-solving heuristic: H.Seq.) on 1000+ variable benchmarks for 50 instances within 60s solving limit.**
>
> | Method | Type | SC(1000,500) Obj.(↓) | SC(1000,500) Node(↓) | SC(1000,500) Time(↓) | SC(1000,500) Dual Gap | CA(1000,500) Obj.(↑) | CA(1000,500) Node(↓) | CA(1000,500) Time(↓) | CA(1000,500) Dual Gap |
> |--------|------|----------------------|----------------------|----------------------|-----------------------|----------------------|----------------------|----------------------|-----------------------|
> | SB | Exact | **230.5** ± 28.3 | 39.0 ± 41.2 | 19.1 ± 15.0 | 0.21 ± 1.23(%) | 146.0 ± 26.6 | 1.0 ± 0.0 | 3.4 ± 1.0 | 0.00 ± 0.00(%) |
> | Rand | Random | 235.9 ± 30.5 | 40.1 ± 45.3 | 20.0 ± 14.9 | 0.21 ± 1.23(%) | 146.0 ± 26.6 | 1.0 ± 0.0 | 3.4 ± 1.0 | 0.00 ± 0.00(%) |
> | GIN | H. Graph | 230.5 ± 28.3 | 38.8 ± 40.0 | 18.6 ± 15.4 | 0.21 ± 1.23(%) | 146.0 ± 26.6 | 1.0 ± 0.0 | 1.8 ± 0.2 | 0.00 ± 0.00(%) |
> | GraphGPS | H. Graph | 230.5 ± 28.3 | 38.9 ± 44.8 | 17.9 ± 15.1 | 0.21 ± 1.23(%) | 146.0 ± 26.6 | 1.0 ± 0.0 | 1.8 ± 0.2 | 0.00 ± 0.00(%) |
> | MILPnet/2 | H. Seq. L=1 | 230.5 ± 28.3 | 37.7 ± 39.8 | 16.5 ± 15.5 | 0.21 ± 1.23(%) | 146.0 ± 26.6 | 1.0 ± 0.0 | 1.7 ± 0.2 | 0.00 ± 0.00(%) |
> | MILPnet/2 | H. Seq. L=2 | 230.5 ± 28.3 | **37.7** ± 39.8 | 16.5 ± 15.6 | 0.21 ± 1.23(%) | **146.0** ± 26.6 | 1.0 ± 0.0 | **1.5** ± 0.2 | 0.00 ± 0.00(%) |
> | MILPnet/2 | H. Seq. L=3 | 230.5 ± 28.3 | 38.5 ± 41.0 | **16.2** ± 15.4 | 0.17 ± 1.21(%) | 146.0 ± 26.6 | 1.0 ± 0.0 | 1.5 ± 0.2 | 0.00 ± 0.00(%) |
> | MILPnet/2 | H. Seq. L=4 | 230.5 ± 28.3 | 38.3 ± 42.0 | 16.3 ± 15.5 | **0.17** ± 1.21(%) | 146.0 ± 26.6 | 1.0 ± 0.0 | 1.5 ± 0.2 | **0.00** ± 0.00(%) |
> | MILPnet/3 | H. Seq. L=1 | 230.5 ± 28.3 | 37.7 ± 39.9 | 16.5 ± 15.5 | 0.21 ± 1.23(%) | 146.0 ± 26.6 | 1.0 ± 0.0 | 1.7 ± 0.2 | 0.00 ± 0.00(%) |
> | MILPnet/4 | H. Seq. L=1 | 230.5 ± 28.3 | 37.7 ± 39.8 | 16.5 ± 15.5 | 0.21 ± 1.23(%) | 146.0 ± 26.6 | 1.0 ± 0.0 | 1.7 ± 0.2 | 0.00 ± 0.00(%) |
>
> The results (Table 9 and Figure 5 in the revised version) show that:
>
>      **(1)** MILPnet, as a pre-solve heuristic, reduces both the number of branching nodes and branching time compared to Full Strong Branching (FSB) and Graph-based pre-solve algorithms.
>
>     **(2)** MILPnet achieves substantially faster dual-gap closure, yielding earlier high-quality incumbent solutions and consistently improved branching time.

---

> ### Author Response · Authors · 2025-11-27
>
> ## Reponse to Weakness 2
>
> We appreciate this insightful comment. Our paper specifically targets the *representation learning layer of MILP* and proposes a sequence-based representation as a substitute for conventional bipartite graph representations. We agree that recent ML-augmented solving frameworks (e.g., ConPAS[2]) provide important complementary perspectives.
>
> To address the reviewer's concern and empirically verify that our approach is compatible with state-of-the-art solver-learning architectures, we conducted new experiments on three very large real-world MILP benchmarks with 10,000+ variables from MIPLIB 2017[3] **(30n20b8: 18380 variables & 576 constraints; blp-ic98: 13640 variables & 717 constraints; blp-ar98: 16021 variables & 1128 constraints)**. Following the ConPAS training methodology, we trained both GCN and MILPnet on Set Cover (SC) problems and evaluated their cross-domain transfer performance on the three heterogeneous benchmarks. Results are provided in Table 10 and Figure 12 (Appendix) in the revised version.
>
>
> **Table 10: Heterogeneous generalization on very-large benchmarks (MIPLIB 2017) within 1500s solving.**
>
> | Method | 30n20b8 (18.4K) Time | 30n20b8 (18.4K) Obj | blp-ic98 (13.6K) Time | blp-ic98 (13.6K) Obj | blp-ar98 (16.0K) Time | blp-ar98 (16.0K) Obj |
> |--------|----------------------|---------------------|----------------------|---------------------|----------------------|---------------------|
> | SCIP | 163.96 | 302.00 | 1500.15 | 4620.13 | 1500.10 | 6215.35 |
> | ConPAS(GCN) | 175.86 | 302.00 | 1500.00 | 4817.66 | 1500.01 | 6254.08 |
> | ConPAS(MILPnet) | **94.64** | **302.00** | **1500.00** | **4588.51** | **1500.01** | **6220.57** |
>
> As shown in Table 10 and Figure 12, MILPnet-based sequence representation within the ConPAS framework outperforms the graph-based ConPAS method on these large benchmarks. Under the same time limits, MILPnet-ConPAS achieves faster convergence to optimal solutions and obtains better objective values. Compared to the state-of-the-art solver SCIP, MILPnet-ConPAS shows competitive performance in both time-to-optimality and solution quality.
>
> While our work focuses on representation learning rather than end-to-end solver design, these results indicate that our sequence representation can be integrated with existing ML-MILP frameworks to improve performance.
>
> ## Reponse to Weakness 3
>
> We have conducted ablation studies on attention depth and maximum scale window size in two settings:
>
> ### (i) Ablation on Foldable instances
>
> We evaluated different attention depths on FOLD(20,\*) and FOLD(50,\*) instances. Results are summarized below (Table 8 and Figure 8 in the revised version).
>
> **Table 8: Ablation study on the number of multi-scale attention blocks (L).**
>
> | Number of blocks | Method + Arch | FOLD(20,16) MSE | FOLD(20,16) ErrorN | FOLD(50,20) MSE | FOLD(50,20) ErrorN |
> |------------------|---------------|-----------------|--------------------|-----------------|--------------------|
> | GCN (Original) | GNN + Graph | 0.3070 | 5000 | 0.4719 | 5000 |
> | L=1 | MILPnet + Seq | 0.0006 | 0 | 0.0003 | 0 |
> | L=2 | MILPnet + Seq | 0.0001 | 0 | 0.0001 | 0 |
> | L=3 | MILPnet + Seq | 0.0004 | 0 | 0.0001 | 0 |
>
> The results show that increasing attention depth can accelerate representation learning in early training, but the final generalization capability converges to similar levels across depths. All tested MILPnet configurations consistently outperform graph-based methods.
>
> ### (ii) Ablation on large real-world instances
>
> We examined both attention depth and maximum scale window size on large real-world benchmarks (see revised Figure 5 and our response to W1&Q1). Under equal training time, different attention depths and maximum scale windows show some impact on solving performance, but the differences are relatively small in the limited time training. Deeper attention blocks tend to tighten the dual gap more rapidly (in SC solving L=2&3&4 can get shorter solving time and smaller dual gap). Across all test configurations, the sequence-based approach consistently outperforms graph-based methods.
>
> These results confirm that MILPnet is robust to hyperparameter choices and that its performance advantage stems from the sequence representation itself rather than a single architectural setting.

---

> ### Author Response · Authors · 2025-11-27
>
> ## Reply to Weakness 4
>
> As described in our response to W2, we conducted cross-domain generalization experiments on very-large real-world benchmarks. Specifically, we trained both GCN and MILPnet on Set Cover (SC) problems with ConPAS pipeline, then transferred the trained models to three completely heterogeneous very large public MILP benchmarks with 10,000+ variables from MIPLIB 2017 **(30n20b8: 18380 variables & 576 constraints; blp-ic98: 13640 variables & 717 constraints; blp-ar98: 16021 variables & 1128 constraints)**.  The results, presented in "Reply to W2", Table 10, and Figure 12 (Appendix), show that MILPnet-based sequence representation achieves stronger cross-domain transfer capability than graph-based alternatives. Under the same time limits, MILPnet-ConPAS reaches dual-convergence faster and achieves smaller objective values than SCIP (obj:2/3,time:3/3) and GCN-ConPAS (obj:3/3,time:3/3). These experiments provide evidence of MILPnet's applicability to real-world industrial problems.
>
> ---
>
> ## Reply to Question 2
>
> Yes, MILPnet is designed to be permutation-invariant to both constraint and variable orderings. Theorem 6 in the original Appendix provides a formal proof of constraint permutation invariance, along with corresponding experimental validation (Table 7 in the original manuscript). This theorem rigorously establishes that the MILP-sequence space is homeomorphic under arbitrary permutations of constraint tokens.
>
> To further strengthen the theoretical foundation, we have now added a new Theorem (Theorem 9, highlighted in blue in the revision) to establish variable permutation invariance, demonstrating that the MILP-sequence space is also homeomorphic under arbitrary permutations of variable coordinates within each token. The complete proof is provided in the revised Appendix (highlighted in blue) and the core statement is:
>
> **Theorem (Variable Permutation Invariance)**
>
> Let $H^{\text{MILP}}$ be the topological space of MILP-sequences defined in Section 3.
>
> For any permutation $\pi$ on variable indices $\{1,2,\dots,n\}$, the MILP-sequence space is homeomorphic under variable permutation:
>
> $$H^{\text{MILP}} \cong H^{\text{MILP}}_\pi,$$
>
> where $H^{\text{MILP}}_\pi$ denotes the space after permuting variable coordinates according to $\pi$.
>
> Consequently, the feasibility mapping $\Phi_{\text{feas}}$, optimal objective value mapping $\Phi_{\text{obj}}$, and optimal solution mapping $\Phi_{\text{solu}}$ are all invariant under variable permutation.
>
> ---
>
> The proof explicitly constructs a variable-coordinate permutation map applied token-wise, and verifies bijectivity and continuity for both the forward and inverse mappings, establishing a homeomorphism. To empirically support this theoretical result, we added new experiments (Table 11 in the revised manuscript).
>
> **Table 11: Experiments on variable permutation invariance.** "V-Or \*" represents randomly permuted variable order in the MILP-sequence. "Original \*" represents the original order.
>
> | Order | Method + Arch | FOLD(20,16) MSE | FOLD(20,16) ErrorN | FOLD(50,20) MSE | FOLD(50,20) ErrorN |
> |-------|---------------|-----------------|--------------------|-----------------|--------------------|
> | GCN (Original) | GNN + Graph | 0.3070 | 5000 | 0.4719 | 5000 |
> | Original | MILPnet + Seq | 0.0003 | 0 | 0.0003 | 0 |
> | V-Or 1 | MILPnet + Seq | 0.0006 | 0 | 0.0001 | 5 |
> | V-Or 2 | MILPnet + Seq | 0.0003 | 0 | 0.0003 | 0 |
> | V-Or 3 | MILPnet + Seq | 0.0004 | 0 | 0.0005 | 3 |
>
> These experiments demonstrate that MILPnet maintains consistent performance under both constraint and variable permutations, confirming that our sequence-based representation does not compromise permutation invariance both theoretically and practically.
>
> In summary, Theorem 6 (constraint permutation invariance) and Theorem 7 (variable permutation invariance), along with the experimental evidence, establish *double permutation invariance* of our framework. Importantly, these results validate that the sequence formulation in MILPnet successfully preserves the inherent permutation invariance properties of MILP problems. Thus, no additional data augmentation or canonicalization is required.
>
> ---
>
> ## Response to Question 3
>
> Thank you for this insightful theoretical question. We have added a comprehensive stability analysis in the new **Appendix H** (highlighted in blue in the revised paper). Our analysis establishes stability properties for the three MILP mappings under small coefficient perturbations. Specifically, we prove that $\Phi_{\text{obj}}$ exhibits strong stability (Lipschitz continuity), while $\Phi_{\text{feas}}$ and $\Phi_{\text{solu}}$ exhibit conditional stability under appropriate non-degeneracy conditions when perturbations are bounded. These theoretical results provide guarantees that MILPnet can maintain stable performance in representing MILP instances despite small perturbations in problem coefficients.

---

> ### Author Response · Authors · 2025-11-27
>
> ## Response to Question 4
>
> Thank you for this practical concern. **Our experiments evaluate MILPnet across a wide range of problem scales and hyperparameters**, and the results indicate that the model is robust to both window-size settings and sequence length.
>
> We evaluated MILPnet on problems ranging from small instances (FOLD(20,6)) to very large instances (blp-ar98 with 16,021 variables and 1,128 constraints). Across all scales, MILPnet consistently outperforms graph-based representations in feasibility classification, optimal solution prediction, warm-start pre-solve heuristics, and representation learning.
>
> Regarding hyperparameter sensitivity, as shown in our responses to W1 and W3, different window sizes and sequence lengths with varying model parameters show that model performance is relatively stable across these choices. **More importantly, across all tested configurations, the sequence-based model maintains performance improvements over graph-based models**.
>
> In practical industrial settings with longer pre-training time, the impact of window size and sequence length diminishes further, while sequence models maintain consistent advantages over graph models. Our experiments from small-scale to industrial-scale problems demonstrate that MILPnet's performance is reasonably robust to hyperparameter choices and sequence lengths.
>
> ---
>
> ## Response to Question 5
>
> Thank you for this inspiring suggestion. We agree that hybridizing high-order graph neural networks with sequence-based representation methods could further enhance the representation capability for MILP instances from both graph and sequence perspectives. By potentially establishing connections between the topological spaces of graph and sequence representations, this approach could achieve stronger representation power. We recognize its value and will pursue it in future research.
>
>
>
> [1] Vinod Nair.Solving Mixed Integer Programs Using Neural Networks,2021.https://arxiv.org/abs/2012.13349
>
> [2] Taoan Huang.Contrastive Predict-and-Search for Mixed Integer Linear Programs,2024.https://openreview.net/forum?id=zatLnLvbs8
>
> [3] Gleixner. MIPLIB 2017: Data-Driven Compilation of the 6th Mixed-Integer Programming Library.https://doi.org/10.1007/s12532-020-00194-3

---

> > ### Comment · Reviewer_DtRt · 2025-11-27
> >
> > Thanks for the authors' response. My concerns are addressed. This is a technically solid paper and I lean towards acceptance.

---

> > > ### Author Response · Authors · 2025-11-28
> > >
> > > Thank you for your expert insights! Your suggestions have genuinely energized our path forward, and we deeply appreciate your support and recognition.

---

### Official Review · Reviewer_qd9C · 2025-10-30

**Soundness:** 3
**Presentation:** 2
**Contribution:** 2
**Rating:** 6
**Confidence:** 4

**Summary:**

This paper proposes MILPnet, a multi-scale hybrid attention framework that departs from traditional graph-based models to address the Mixed Integer Linear Programming problem. Instead of modeling MILPs as bipartite graphs, MILPnet represents each MILP instance as a geometric feature sequence. The paper proves that MILPnet can approximate MILP feasibility, optimal objective value, and optimal solution mappings with arbitrarily small error in a measurable topological space, and empirical results demonstrate that it outperforms graph-based methods by multiple orders of magnitude in feasibility prediction accuracy and convergence speed on Foldable MILPs.

**Strengths:**

1) By modeling MILPs as geometric feature sequences, it overcomes the Weisfeiler-Lehman test’s expressive limit, enabling effective distinction of Foldable MILP instances that graph-based models fail to differentiate.

2) It provides strict mathematical proofs that guarantee arbitrary-precision approximation of MILP feasibility, optimal objective value, and optimal solution mappings in a measurable topological space, ensuring reliability.

3) Empirically, it outperforms graph-based baselines by multiple orders of magnitude in feasibility prediction accuracy and convergence speed on Foldable MILPs.

**Weaknesses:**

1. The maximum window size in its multi-scale attention mechanism does not follow a monotonic pattern—smaller windows accelerate convergence on simple instances (e.g., FOLD(20,6)) but degrade performance on complex ones (e.g., FOLD(100,20)), requiring careful tuning for different problem scales.

2) The paper provides limited empirical validation on extremely large-scale MILP instances, leaving uncertainty about its scalability in industrial-grade scenarios with massive constraints/variables.

3) Although padding ensures topological equivalence, the framework’s performance on MILP instances with drastically varying constraint counts lacks in-depth analysis, raising questions about its stability across highly heterogeneous problem sizes.

**Questions:**

Please see the Weaknesses.

---

> ### Author Response · Authors · 2025-11-27
>
> We sincerely thank the reviewer for these insightful observations, which have motivated us to conduct additional experiments and provide deeper analysis. All revisions have been highlighted in blue in the updated manuscript.
>
> ---
>
> ## Response to Weakness 1 (Window Size Tuning)
>
> We thank the reviewer for the thoughtful observation. We would like to clarify that the observed need for window size tuning is primarily a consequence of *limited training time and computational resources*. Specifically, Figure 6b in the original paper shows results under a tight 5-minute pre-training constraint. However, as evidenced in Figure 6 in the appendix, when training time is sufficient, the impact of maximum window size on MILPnet's performance diminishes significantly. These results suggest that:
>
> **(1)** Under resource-constrained applications (e.g., limited training time or GPU availability), window size selection indeed affects convergence speed.
>
> **(2)** In practical deployments with adequate training resources, the sensitivity to window size becomes negligible, as the model has sufficient capacity to learn effective representations across different window configurations.
>
> Therefore, window size tuning is primarily relevant for rapid prototyping or resource-limited settings, rather than a fundamental limitation of the proposed framework.
>
> ---

---

> ### Author Response · Authors · 2025-11-27
>
> ## Response to Weakness 2 (Large-Scale Validation)
>
> We thank the reviewer for this professional suggestion. In response, we have conducted extensive additional experiments on large-scale and extremely large-scale benchmarks.
>
> ### (W2.1) Experiments on large public benchmarks (1000 variables)
>
> We evaluated a 1-hour pretrained MILPnet in a neural diving-style heuristic presolver that constructs trust-region subproblems integrated with branch-and-bound, comparing against graph-based networks and strong branching. We report the number of branch-and-bound nodes, solve time, dual gap, and final objective value.
>
> The table below is the revised Table 9, where "/w" denotes the maximum scale size of MILPnet and "L" denotes the depth of the multi-scale attention mechanism.
>
> **Table: Branch and Bound performance of MILPnet (Neural Diving based pre-solving heuristic: H.Seq.) on 1000+ variable benchmarks for 50 instances within 60s solving limit.**
>
> | Method | Type | SC(1000,500) Obj.(↓) | SC(1000,500) Node(↓) | SC(1000,500) Time(↓) | SC(1000,500) Dual Gap | CA(1000,500) Obj.(↑) | CA(1000,500) Node(↓) | CA(1000,500) Time(↓) | CA(1000,500) Dual Gap |
> |--------|------|----------------------|----------------------|----------------------|-----------------------|----------------------|----------------------|----------------------|-----------------------|
> | SB | Exact | **230.5** ± 28.3 | 39.0 ± 41.2 | 19.1 ± 15.0 | 0.21 ± 1.23(%) | 146.0 ± 26.6 | 1.0 ± 0.0 | 3.4 ± 1.0 | 0.00 ± 0.00(%) |
> | Rand | Random | 235.9 ± 30.5 | 40.1 ± 45.3 | 20.0 ± 14.9 | 0.21 ± 1.23(%) | 146.0 ± 26.6 | 1.0 ± 0.0 | 3.4 ± 1.0 | 0.00 ± 0.00(%) |
> | GIN | H. Graph | 230.5 ± 28.3 | 38.8 ± 40.0 | 18.6 ± 15.4 | 0.21 ± 1.23(%) | 146.0 ± 26.6 | 1.0 ± 0.0 | 1.8 ± 0.2 | 0.00 ± 0.00(%) |
> | GraphGPS | H. Graph | 230.5 ± 28.3 | 38.9 ± 44.8 | 17.9 ± 15.1 | 0.21 ± 1.23(%) | 146.0 ± 26.6 | 1.0 ± 0.0 | 1.8 ± 0.2 | 0.00 ± 0.00(%) |
> | MILPnet/2 | H. Seq. L=1 | 230.5 ± 28.3 | 37.7 ± 39.8 | 16.5 ± 15.5 | 0.21 ± 1.23(%) | 146.0 ± 26.6 | 1.0 ± 0.0 | 1.7 ± 0.2 | 0.00 ± 0.00(%) |
> | MILPnet/2 | H. Seq. L=2 | 230.5 ± 28.3 | **37.7** ± 39.8 | 16.5 ± 15.6 | 0.21 ± 1.23(%) | **146.0** ± 26.6 | 1.0 ± 0.0 | **1.5** ± 0.2 | 0.00 ± 0.00(%) |
> | MILPnet/2 | H. Seq. L=3 | 230.5 ± 28.3 | 38.5 ± 41.0 | **16.2** ± 15.4 | 0.17 ± 1.21(%) | 146.0 ± 26.6 | 1.0 ± 0.0 | 1.5 ± 0.2 | 0.00 ± 0.00(%) |
> | MILPnet/2 | H. Seq. L=4 | 230.5 ± 28.3 | 38.3 ± 42.0 | 16.3 ± 15.5 | **0.17** ± 1.21(%) | 146.0 ± 26.6 | 1.0 ± 0.0 | 1.5 ± 0.2 | **0.00** ± 0.00(%) |
> | MILPnet/3 | H. Seq. L=1 | 230.5 ± 28.3 | 37.7 ± 39.9 | 16.5 ± 15.5 | 0.21 ± 1.23(%) | 146.0 ± 26.6 | 1.0 ± 0.0 | 1.7 ± 0.2 | 0.00 ± 0.00(%) |
> | MILPnet/4 | H. Seq. L=1 | 230.5 ± 28.3 | 37.7 ± 39.8 | 16.5 ± 15.5 | 0.21 ± 1.23(%) | 146.0 ± 26.6 | 1.0 ± 0.0 | 1.7 ± 0.2 | 0.00 ± 0.00(%) |
>
> The results (Table 9 and Figure 5 in the revised version) show that:
>
> **(1)** MILPnet significantly reduces both node count and solve time compared to strong branching.
>
> **(2)** MILPnet further improves the dual gap compared to graph-based methods.
>
> ### (W2.2) Experiments on extremely large-scale real-world benchmarks (10,000+ variables)
>
> To test scalability under realistic industrial settings, we further evaluated MILPnet on three extremely large MILP instances from MIPLIB 2017 (30n20b8: 18380 variables & 576 constraints; blp-ic98: 13640 variables & 717 constraints; blp-ar98: 16021 variables & 1128 constraints). Notably, following the ConPAS training methodology, we trained both GCN and MILPnet on Set Cover (SC) problems and then transferred them directly to the three heterogeneous benchmarks without retraining.
>
> **Table: Heterogeneous generalization on very-large benchmarks (MIPLIB 2017) within 1500s solving.**
>
> | Method | 30n20b8 (18.4K) Time | 30n20b8 (18.4K) Obj | blp-ic98 (13.6K) Time | blp-ic98 (13.6K) Obj | blp-ar98 (16.0K) Time | blp-ar98 (16.0K) Obj |
> |--------|----------------------|---------------------|----------------------|---------------------|----------------------|---------------------|
> | SCIP | 163.96 | 302.00 | 1500.15 | 4620.13 | 1500.10 | 6215.35 |
> | ConPAS(GCN) | 175.86 | 302.00 | 1500.00 | 4817.66 | 1500.01 | 6254.08 |
> | ConPAS(MILPnet) | **94.64** | **302.00** | **1500.00** | **4588.51** | **1500.01** | **6220.57** |
>
> Results (Table 10 and Figure 12 in the revised version) show that:
>
> **(1)** Even on extremely large MILP benchmarks, the MILPnet-based ConPAS consistently outperforms the graph-based ConPAS and SCIP.
>
> **(2)** Under the same time limits, MILPnet achieves faster dual convergence.
>
> The results provide compelling evidence that MILPnet scales effectively to industrial-grade MILP problem sizes. Moreover, they empirically demonstrate MILPnet's strong *cross-domain transfer* capability, successfully generalizing from SC to completely heterogeneous large-scale benchmarks.

---

> ### Author Response · Authors · 2025-11-27
>
> ## Response to Weakness 3 (Stability)
>
> We thank the reviewer for highlighting this important point. Our experiments and theory jointly provide compelling evidence that the proposed framework remains stable across MILP instances with drastically varying constraint counts.
>
> ### (W3.1) Performance across diverse constraint counts
>
> Our sequence representation and MILPnet architecture consistently outperform graph-based methods across a broad range of problem sizes, from 6 constraints up to 1,128 constraints(10~1,132sequence length), demonstrating **fewer required parameters, faster inference speed, and higher solution quality**. This indicates that the homeomorphism induced by our padding strategy maintains its effectiveness from small to very large scales.
>
> ### (W3.2) Cross-domain transfer on heterogeneous problems
>
> The new experiments reported in our response to Weakness 2 (Table 10 and Figure 12 in the revised version) further validate stability across heterogeneous problems:
>
> **(1)** Models trained on Set Cover successfully transfer to completely different, extremely large, real-world MILP instances, despite dramatic differences in constraint structure, variable counts, and problem semantics.
>
> **(2)** Performance remains strong even when test instances are orders of magnitude larger than training instances.
>
> This confirms that the framework is not sensitive to variation in constraint counts or problem structure.
>
> **(3)** Theoretical justification
>
> The empirical findings are supported by our theoretical guarantees:
>
>      **(I)** Theorem 4 ensures that padding preserves the topological structure.
>
>      **(II)** The approximation corollary on infinite datasets establishes that MILPnet retains approximation guarantees over compact infinite sets under $\mu_{H^{\text{MILP}}}$, providing theoretical assurance of stability across heterogeneous problem scales.
>
> ---
>
> ## (W3.3) Summary
>
> In summary, our comprehensive evaluation, including small-scale (6 constraints), medium/large-scale (100–1,000 variables), and extremely large-scale (10,000+ variables) problems, along with cross-domain transfer experiments, demonstrates that MILPnet maintains robust and stable performance across highly heterogeneous problem sizes. Both empirical evidence and theoretical guarantees support that the padding-based topological equivalence preserves stability rather than introducing sensitivity to constraint-count variation.
>
> [1] Taoan Huang.Contrastive Predict-and-Search for Mixed Integer Linear Programs,2024.https://openreview.net/forum?id=zatLnLvbs8
>
> [2] Gleixner. MIPLIB 2017: Data-Driven Compilation of the 6th Mixed-Integer Programming Library.https://doi.org/10.1007/s12532-020-00194-3

---

### Official Review · Reviewer_vnbQ · 2025-11-02

**Soundness:** 2
**Presentation:** 2
**Contribution:** 2
**Rating:** 4
**Confidence:** 3

**Summary:**

This paper proposes MILPnet, a multi-scale hybrid attention model that represents MILP problems as geometric sequences instead of graphs to address GNN limitations on "Foldable MILPs." It encodes constraints, objectives, and variables into sequences, uses shifted-window attention for local-global feature capture, and proves approximation capabilities for feasibility, optimal values, and solutions.

**Strengths:**

The paper shifts from graph-based to sequence-based representations for MILPs, addressing GNN limits in expressiveness for Foldable instances. It provides proofs of universal approximation results. The numerical results report improvements in accuracy and efficiency on Foldable instances.

**Weaknesses:**

1. In Theorem 2-4, only a finite dataset $D$ is considered. Since the dataset is finite, I do not think the notation $P$ is needed and the $\epsilon$ in equations (9) and (10) is not needed. The authors are basically borrowing proof ideas from the work of Chen et al but the proof presentation and notation have much room for improvement.
2. The sequential representation breaks the permutation-invariance property of MILP.

**Questions:**

None

---

> ### Author Response · Authors · 2025-11-27
>
> ## Response to Weakness 1
> We sincerely thank the reviewer for the thorough and professional evaluation. All revisions have been highlighted in blue in the updated manuscript.
> ### (W1.1) Regarding extensions to infinite sets:
>
> We would like to clarify that the original submission already included corollary proofs for the approximation of feasibility and optimal solution mappings on infinite sets in the original Appendix (Appendix G in newest), which were explicitly referenced in the main text. To address the reviewer's concern more comprehensively, we have now added a complete corollary proof for the optimal objective value mapping on infinite sets: (see Corollary 2 in the revised Appendix G, highlighted in blue).
>
> **Corollary (Extension to Compact Infinite Set)**
>
> Let $K \subset H^{\text{MILP}}$ be a compact subset equipped with the measure $\mu_{H^{\text{MILP}}}$ defined in main text. Define the probability measure $P$ on $K$ by normalization: $P(A) = \frac{\mu_{H^{\text{MILP}}}(A)}{\mu_{H^{\text{MILP}}}(K)}$ for any measurable set $A \subset K$.
>
> For any $\epsilon, \delta > 0$, there exist two neural networks $F_{\text{HYA},1}, F_{\text{HYA},2} \in \mathcal{F}_{\text{HYA}}^{\text{MILPnet}}$ such that:
>
> **1. Classification of finite objective values:**
>
> $$P\left(\mathbb{I}\_{F\_{\text{HYA},1}(x) > \frac{1}{2}} \neq \mathbb{I}\_{\Phi_{\text{obj}}(x) \in \mathbb{R}}\right) < \epsilon$$
>
> **2. Regression of objective values:**
>
> $$P\left(\left|F_{\text{HYA},2}(x) - \Phi_{\text{obj}}(x)\right| > \delta\right) < \epsilon, \quad \forall x \in K \cap \Phi_{\text{obj}}^{-1}(\mathbb{R})$$
>
> ---
>
> **Proof:**
>
> The proof parallels that of Theorem 3 in our main text.
>
> **1. Classification Problem.**
>
> Since $\Phi_{\text{obj}}: H^{\text{MILP}} \to \mathbb{R} \cup \{\infty\}$ is measurable and $K$ is compact with $\mu_{H^{\text{MILP}}}(K) < \infty$, we apply Lusin's Theorem.
>
> For any $\epsilon > 0$, there exists a closed set $C \subset K$ such that:
> $$P(K \setminus C) = \frac{\mu_{H^{\text{MILP}}}(K \setminus C)}{\mu_{H^{\text{MILP}}}(K)} < \frac{\epsilon}{2}$$
>
> and  $\Phi_{\text{obj}}|_C$
>
> is continuous.  By the Universal Approximation Theorem, there exists
> $F_{\text{HYA},1} \in \mathcal{F}_{\text{HYA}}^{\text{MILPnet}}$
>
> such that
>
> $\sup\_{x \in C} \left|F\_{\text{HYA},1}(x) - \mathbb{I}\_{\Phi_{\text{obj}}(x) \in \mathbb{R}}\right| < \frac{1}{4}$
>
> which guarantees that for all $x \in C$:
>
> $$\mathbb{I}\_{F\_{\text{HYA},1}(x) > \frac{1}{2}} = \mathbb{I}\_{\Phi\_{\text{obj}}(x) \in \mathbb{R}}$$
>
> The set of misclassified points is contained in $K \setminus C$, therefore:
>
> $P\left(\mathbb{I}\_{F\_{\text{HYA},1}(x) > \frac{1}{2}} \neq \mathbb{I}\_{\Phi\_{\text{obj}}(x) \in \mathbb{R}}\right) \leq P(K \setminus C) < \epsilon$
>
> **2. Regression Problem.**
>
> Let $K_{\text{finite}} = K \cap \Phi_{\text{obj}}^{-1}(\mathbb{R})$. Define the conditional probability measure on $K_{\text{finite}}$ by:
>
> $$P_{\text{finite}}(A) = \frac{\mu_{H^{\text{MILP}}}(A)}{\mu_{H^{\text{MILP}}}(K_{\text{finite}})}, \quad A \subset K_{\text{finite}}$$
>
> By applying Lusin's Theorem to $\Phi_{\text{obj}}$ on $K_{\text{finite}}$, for any $\epsilon > 0$, there exists a closed set $C' \subset K_{\text{finite}}$ such that:
>
> $$P_{\text{finite}}(K_{\text{finite}} \setminus C') < \frac{\epsilon}{2}$$
> and $\Phi_{\text{obj}}|_{C'}$
>
> is continuous.
> By the Universal Approximation Theorem, there exists $F_{\text{HYA},2} \in \mathcal{F}_{\text{HYA}}^{\text{MILPnet}}$
>
> such that:
>
> $$\sup_{x \in C'} \left|F\_{\text{HYA},2}(x) - \Phi\_{\text{obj}}(x)\right| < \delta$$
>
> Therefore:
>
> $$P\_{\text{finite}}\left(\left|F_{\text{HYA},2}(x) - \Phi\_{\text{obj}}(x)\right| > \delta\right) \leq P\_{\text{finite}}(K\_{\text{finite}} \setminus C') < \epsilon$$
>
>
> ---
> The extended results establish that our approximation guarantees naturally extend from finite datasets to compact infinite subsets of $H^{\text{MILP}}$ under the normalized measure $\mu_{H^{\text{MILP}}}$.
>
> ### (W1.2) Regarding proof presentation and notation for Theorems:
> We thank the reviewer for this insightful suggestion. To enhance clarity, we have revised the theorem statements to articulate the role of probability explicitly. Specifically, we have added  **Remark G.1** in the revised **Appendix G** to clarify that $P$ denotes the probability over the random training process (including initialization and SGD noise), consistent with conventions in machine learning theory.
> Alternatively, if the reviewer prefers an empirical error formulation analogous to $\Phi_{\text{feas}}$, the approximation condition can be equivalently written as (e.g., for the feasibility mapping $\Phi_{\text{feas}}$):
>
> $\frac{1}{|D|} | \\{ x \in D: I_{F_{\text{HYA}}(x)> \frac{1}{2}} \neq \Phi\_{\text{feas}}(x) \\}| < \epsilon$
>
> However, we believe the probabilistic formulation more naturally reflects the stochastic nature of neural network training and aligns with standard notation in machine learning theory.

---

> ### Author Response · Authors · 2025-11-27
>
> ### (W1.3) Regarding proof presentations:
>
> Thank you for your careful review. In the revised version, we have improved the flow and exposition of the proofs (highlighted in blue in the revision). These changes substantially improve clarity while preserving mathematical rigor, and we will incorporate your suggestions to make the presentation more concise.
>
> ---
>
> ## Response to Weakness 2
>
> We sincerely thank the reviewer for the professional and constructive feedback.
>
> ### (W2.1) Constraint permutation invariance:
>
> We would like to respectfully clarify that the original submission already included a formal proof of constraint permutation invariance (Theorem 6 in the original manuscript), along with corresponding experimental validation (Table 7 in the original manuscript). This theorem rigorously establishes that the MILP-sequence space is homeomorphic under arbitrary permutations of constraint tokens.
>
> ### (W2.2) Variable permutation invariance:
>
> To further strengthen the theoretical foundation, we have now added a new Theorem (highlighted in blue in the revision, Theorem 9 in newest version) to establish variable permutation invariance, demonstrating that the MILP-sequence space is also homeomorphic under arbitrary permutations of variable coordinates within each token. The complete proof is provided in the revised Appendix E (highlighted in blue) and the core statement is:
>
> **Theorem (Variable Permutation Invariance)**
>
> Let $H^{\text{MILP}}$ be the topological space of MILP-sequences defined in Section.
>
> For any permutation $\pi$ of variable indices $\{1,2,\dots,n\}$, the MILP-sequence space is homeomorphic under variable permutation:
>
> $$H^{\text{MILP}} \cong H^{\text{MILP}}_\pi,$$
>
> where $H^{\text{MILP}}_\pi$ denotes the space after permuting variable coordinates according to $\pi$.
>
> Consequently, the feasibility mapping $\Phi_{\text{feas}}$, optimal objective value mapping $\Phi_{\text{obj}}$, and optimal solution mapping $\Phi_{\text{solu}}$ are all invariant under variable permutations.
>
> ---
>
> To empirically support this theoretical result, we added new experiments (Table in the revised Appendix B.3).
>
> **Table: Experiments on variable permutation invariance.** "V-Or *" represents randomly permutated variable order in the MILP-sequence. "Original *" represents the original order.
>
> | Order | Method + Arch | FOLD(20,16) MSE | FOLD(20,16) ErrorN | FOLD(50,20) MSE | FOLD(50,20) ErrorN |
> |-------|---------------|-----------------|--------------------|-----------------|--------------------|
> | GCN (Original) | GNN + Graph | 0.3070 | 5000 | 0.4719 | 5000 |
> | Original | MILPnet + Seq | 0.0003 | 0 | 0.0003 | 0 |
> | V-Or 1 | MILPnet + Seq | 0.0006 | 0 | 0.0001 | 5 |
> | V-Or 2 | MILPnet + Seq | 0.0003 | 0 | 0.0003 | 0 |
> | V-Or 3 | MILPnet + Seq | 0.0004 | 0 | 0.0005 | 3 |
>
> These experiments demonstrate that MILPnet maintains consistent performance under both constraint and variable permutations, confirming that our sequence-based representation does not compromise permutation invariance both theoretically and practically.

---

> ### Author Response · Authors · 2025-11-27
>
> ### (W2.3) Summary
>
> Theorems (constraint permutation invariance) and (variable permutation invariance), along with the experimental evidence, establish *double permutation invariance* of our framework. Importantly, these results validate that the sequence formulation in MILPnet successfully preserves the inherent permutation invariance properties of MILP problems.

---

### Meta-Review · Area_Chair_F5QX · 2026-01-02

**Summary:**

The paper proposes MILPnet, a sequence-based attention framework for Mixed Integer Linear Programming (MILP) aimed at overcoming the expressiveness limitations of GNN-based methods (specifically regarding the Weisfeiler-Lehman test and Foldable instances). While the reviewers universally appreciated the novelty of shifting from graph to sequence representations and the theoretical motivation regarding expressiveness, significant concerns regarding empirical validation, baseline comparisons, and representational invariance render it marginally above the threshold of acceptantce at this time.

**Reviewer Concerns:**

1. Lack of Comprehensive Baselines and Generalization: A primary concern raised by Reviewer DtRt is the narrow scope of the comparative analysis. While the paper compares against standard GNN representations, it fails to benchmark against state-of-the-art ML-augmented Branch-and-Bound (B&B) methods (e.g., learned branching/cuts), RL-based solvers, or modern reduction frameworks (e.g., ConPaS). Furthermore, Reviewer qd9C noted that the validation is heavily skewed toward "Foldable" instances, with insufficient evidence of scalability to large-scale, industrial-grade instances or general MILP sets (e.g., MIPLIB). The consensus is that the empirical case for "universality" is currently overstated without broader generalization tests.

2. Permutation Invariance and Representation: Both Reviewers vnbQ and DtRt highlighted a critical theoretical disconnect: MILP constraints and variables are inherently permutation-invariant sets, yet the proposed sequence-based modeling breaks this property. The paper does not sufficiently address how this ordering bias is mitigated (e.g., via data augmentation or canonicalization), raising doubts about the robustness of the approach when constraint ordering changes.

3. Theoretical and Notation Rigor: Reviewer vnbQ pointed out technical flaws in the presentation of the proofs (Theorems 2-4), specifically questioning the notation regarding finite datasets versus topological spaces. The reviewer suggested that the proof presentation borrows heavily from existing work (Chen et al.) without sufficient precision or adaptation to the specific claims made here.

4. Ablation and Hyperparameter Sensitivity Reviewers: qd9C and DtRt found the method sensitive to hyperparameters, particularly the multi-scale attention window sizes, which appear to lack a monotonic performance pattern (degrading on complex instances if not carefully tuned). Additionally, the lack of ablation studies on feature choices, attention depth, and varying MILP families makes it difficult to isolate the source of the performance gains.

Most of the concerns were addressed during the rebuttal process. However, I still have reservations about the paper's core concept—specifically, the attention framework that models Mixed Integer Linear Programming (MILP) problems as geometric sequences rather than graphs.

**Reviewer Scores:**

Reviewer vnbQ might maintain his/her original score. Reviewer DtRt and qd9C have already adjusted their score to 6.

---

### Decision · Program_Chairs · 2026-01-26

Accept (Poster)